

# Time-reversal-broken Weyl semimetal in the Hofstadter regime

**Faruk Abdulla[1⋆], Ankur Das[2†], Sumathi Rao[1,3] and Ganpathy Murthy[4]**

**1** Harish-Chandra Research Institute, HBNI, Chhatnag Road, Jhunsi, Allahabad, India
**2** Department of Condensed Matter Physics, Weizmann Institute of Science, Israel
**3** International Centre for Theoretical Sciences, Tata Institute of Fundamental Research, Bengaluru, India
**4** Department of Physics and Astronomy, University of Kentucky, Lexington, KY, USA

⋆ farukabdulla@hri.res.in, † ankur.das@weizmann.ac.il

## Abstract

We study the phase diagram for a lattice model of a time-reversal-broken three-dimensional Weyl semimetal (WSM) in an orbital magnetic field $B$ with a flux of $p/q$ per unit cell ($0 \le p \le q-1$), with minimal crystalline symmetry. We find several interesting phases: (i) WSM phases with $2q$, $4q$, $6q$, and $8q$ Weyl nodes and corresponding surface Fermi arcs, (ii) a layered Chern insulating (LCI) phase, gapped in the bulk, but with gapless surface states, (iii) a phase in which some bulk bands are gapless with Weyl nodes, coexisting with others that are gapped but topologically nontrivial, adiabatically connected to an LCI phase, (iv) a new gapped trivially insulating phase (I′) with (non-topological) counter-propagating surface states, which could be gapped out in the absence of crystal symmetries. Importantly, we are able to obtain the phase boundaries analytically for all $p, q$. Analyzing the gaps for $p = 1$ and very large $q$ enables us to smoothly take the zero-field limit, even though the phase diagrams look ostensibly very different for $q = 1, B = 0$, and $q \to \infty, B \to 0$.

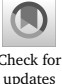

# 1 Introduction

Weyl semimetals (WSM) [1–11] are examples of topological quantum matter which are not fully insulating in the bulk but have an even number of points in the Brillouin zone called Weyl nodes where the conduction and valence bands touch each other. Each Weyl node has a topological charge and acts as a source or sink of the Berry phase in momentum space. The gapless surface states of the WSM consist of Fermi arcs in the surface Brillouin zone (SBZ) which join the projections of the Weyl nodes onto the SBZ.

It is well-known [12–15] that in the presence of parallel external electric and magnetic fields, the density of electrons at individual nodes is not conserved and transport in the WSM is anomalous due to the chiral anomaly. This causes the electrons to be pumped from one Weyl node to another with opposite topological charge until this process is balanced by internode scattering. The surface Fermi arcs also show interesting quantum oscillations [16–19] in an applied magnetic field. Semiclassically, the Lorentz force causes the electrons to move along the Fermi arc, tunnel through the bulk at the Weyl node and then complete the circuit via the Fermi arc on the other surface and tunneling back through the bulk.

Beyond the semiclassical limit, the effect of orbital magnetic fields $B$ on the WSM has been well-studied [12,15] in the continuum limit (magnetic length $\ell = \sqrt{\hbar/eB}$ is much larger than the lattice spacing $a$), where the $B$ field is weak enough that attention can be restricted to states very close to the Weyl nodes. Coming to strong fields, Roy and co-workers [20] have characterized the Hofstadter-Weyl butterfly in a simple two-band model with two Weyl nodes, where the magnetic field is applied parallel to the separation between the Weyl nodes. There has also been some work [21] on obtaining the energy spectra for the Hofstadter Hamiltonians for Weyl and double-Weyl semimetals, where the systems are shown to exhibit 3D quantum Hall effect [18,22,23] for appropriate hopping parameters and rational fluxes.

We go beyond previous work in two ways. First, we consider a two-band model with a

mimimal crystalline symmetries, - i.e., with fully anisotropic hoppings. Here, even without an external magnetic field, we find that there are several new phases of WSMs with 2, 4, 6, and 8 Weyl nodes. Earlier studies [23–26] had imposed crystalline symmetries and had uncovered the WSM phases with 2 and 4 nodes with the LCI phase (also sometimes called a 3D Quantum Hall state [24] or a 3D Chern Insulator [25]), but the phases with 6 and 8 nodes had not been seen earlier and requires unconstrained anisotropic hoppings. Second, unlike earlier studies of the effect of magnetic fields on this model, we consider the case when the external $B$ field is perpendicular to the separation between the nodes. Constraining one of the hopping parameters, we are able obtain the phase boundaries analytically at arbitrary values of $p, q$. Last, but not least, we smoothly connect our results for small but commensurate fields to the semiclassical limit.

Our central result is the set of phase diagrams in Fig. 1 for each $q$ (where $p/q$ quanta of flux go through each unit cell). The phase diagrams are universal in two distinct ways: (i) For each value of $q$, the set of phases and the regions they occupy in parameter space are independent of $p$. (ii) For all $q \geq 1$, phases with $2q$, $4q$, $6q$, and $8q$ Weyl nodes appear in the phase diagram, as well as the trivial insulator and the LCI phase. For all $q \geq 2$, a phase in which gapless Fermi arcs coexist with gapless chiral states spanning the surface BZ (characteristic of the LCI) always appears. For all $q \geq 3$ a topologically trivial insulator with gapless surface modes on certain surfaces (protected by translation symmetry) always appears.

In a particular gauge, the $q$-fold increase in the number of Weyl nodes is essentially a consequence of the $q$-fold increase in the periodicity along the $y$ axis. This $q$-fold increase in periodicity also leads to an increase in the number of phase boundaries and consequently an increase in the number of times each phase appears in the phase diagram. Transitions between phases with differing numbers of Weyl nodes occur via the creation/annihilation of pairs of Weyl nodes of opposite topological charge. The bulk dispersion at the location of the phase transition is quadratic in one direction but linear in the other two. The corresponding surface spectrum of the Fermi arcs is also quadratic in one direction but linear in the other at the phase transitions.

While the phase diagram is independent of $p$, the spectrum strongly depends on $p$. For a certain restricted, but still nontrivial, set of parameters, we can solve for the phase boundaries analytically, which enables us to go to very large values of $q$, where the phase diagram in most of the parameter space approaches a limit as $q \to \infty$. The exception is a complicated patchwork of many phases occurring in a narrow band of parameters, whose width vanishes as $q \to \infty$ (Fig. 8). At first glance, it is puzzling that the phase diagram as $p = 1, q \to \infty$ (Fig. 8) which is the semiclassical limit $B \to 0$, looks very different from that at at $B = 0$ (Fig. 1a). However, a detailed examination of the asymptotic behavior of the gap near zero energy enables us to make the correspondence between the $p = 1, q \to \infty$ and the $B = 0$ phase diagrams (studied in Figs. 8-10).

Many of the features that we find in our analysis can be translated to experimental detection of the new phases via their topological responses. For instance, phases with co-existence of the LCI and a WSM can lead to both of them contributing to the Hall conductance (Figs. 3-5). The Hall conductance can also be tuned by changing parameters to go between phases.

The plan of the paper is as follows. In Sec. 2, we introduce the fully anisotropic two-band lattice model of a WSM with broken time-reversal symmetry and obtain its phase diagram at zero flux. In Sec. 3 we obtain the phase boundaries analytically for $p/q$ flux quanta piercing each unit cell in a direction perpendicular to the Weyl-node separation (the $x$ direction). In Sec. 4 we study the bulk and the surface spectra, paying particular attention to gapless surface states. In Sec. 5 we discuss the weak-field limit in detail and make contact with the continuum description. We conclude in Sec. 6 with a summary, potential caveats in our results, the effects of disorder, and possible directions for the future. A number of straightforward mathematical

details are relegated to the appendices.

## 2 The lattice model and its phase diagram at zero flux

We begin with a time-reversal-broken two-band lattice model [23] of Weyl semi-metal defined on a cubic lattice given by the following Hamiltonian

$$H = \sum_{\mathbf{n},j} c^\dagger(\mathbf{n}) 2M\sigma_x c(\mathbf{n}) - \left( c^\dagger(\mathbf{n} + a\hat{e}_j) \, T_j \, c(\mathbf{n}) + H.c. \right), \tag{1}$$

where $\mathbf{n} = a(n_x, n_y, n_z)$, $n_i$ being integers, denote the lattice sites, $\hat{e}_j$ is the unit vector along $j^{th}$ direction, the $\sigma_i$'s are the Pauli matrices representing the (pseudo)spin and $\mathbf{c}(\mathbf{n}) = (c_\uparrow(\mathbf{n}), c_\downarrow(\mathbf{n}))^T$ are the two-component fermions. Microscopically, the pseudospin label arises from spin-orbit coupling, leading to eigenstates which are linear combinations of spin and orbital eigenstates. The hopping matrices $T_j$, $j = (x, y, z)$, are given by: $T_x = t_x \sigma_x$, $T_y = t_y^{(1)} \sigma_x + i t_y^{(2)} \sigma_y$ and $T_z = t_z^{(1)} \sigma_x + i t_z^{(2)} \sigma_z$. The lattice constant $a$ is set to be unity for the rest of the paper. This model has been studied earlier for isotropic hoppings $t_x = t_y^{(1)} = t_z^{(1)} = t_1$, $t_y^{(2)} = t_z^{(2)} = t_2$ and with $t_1 = t_2$ and is known [23–26] to have a Weyl semimetal (WSM) phase for $|M/t_1| < 3$. In this paper we work with fully anisotropic hoppings, where the model has minimal rotational symmetry - a single two-fold rotation about the $x$-axis (the full symmetry analysis is carried out in the Appendix A).

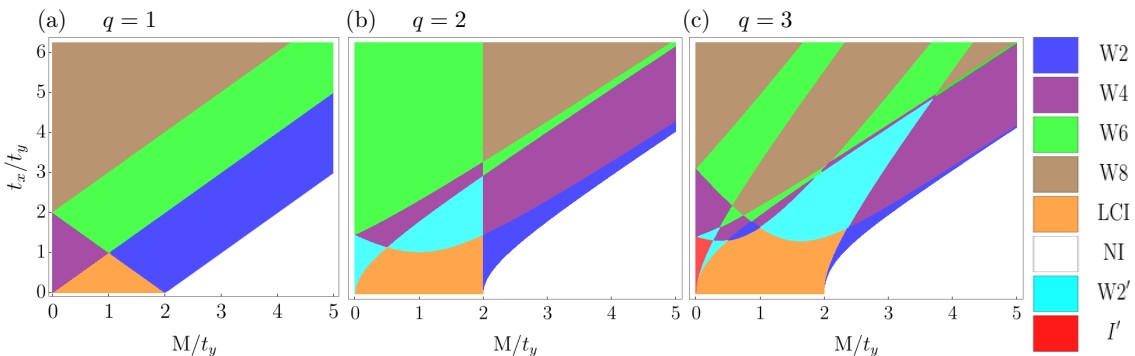

Figure 1: The phase diagram for (a) zero field ($q = 1$) and (b)-(c) for finite commensurate fields ($q = 2, 3$), with $t_z^{(1)} = t_y^{(1)} = t_y$. (a) The gapless phases consisting of WSM phases with 1, 2, 3, and 4 pairs of Weyl nodes are shown in blue, purple, green, and brown. The gapped phase (in orange) is a layered Chern insulator (LCI), where the 2D Chern layers are coupled via $t_x$. The white region at the right bottom corner represents a trivial insulator. In (b)-(c) the WSM phases have $2q$, $4q$, $6q$, and $8q$ Weyl nodes in the bulk, labeled W2, W4, W6, and W8 respectively. In non-zero magnetic fields, two additional phases appear which are labeled W2$'$ and I$'$. The W2$'$ phase (aqua) is gapless and has $2q$ Weyl nodes, but coexists with an LCI, and hence possesses gapless surface modes in addition to the usual Fermi arcs. The I$'$ phase is a gapped, topologically trivial, insulating phase but has counterpropagating zero-energy surface states. We also note that the gapped LCI and I$'$ phases take up regions from the WSM phases and expand with increasing $q$. We emphasize that the phases of the same color in (a), (b), and (c) are not identical. The number of Weyl nodes in corresponding phases is related by the factor $q$.

To set the stage for our nonzero flux results, we first analyse the model at zero flux in the

full parameter space. We find that the topological phase diagram, in general, depends only on three of the hoppings $(t_x, t_y^{(1)}, t_z^{(1)})$ and the onsite mass parameter $M$, which are all real. The Bloch Hamiltonian in **k**-space is

$$h(\mathbf{k}) = 2\left(M - t_x \cos k_x - t_y^{(1)} \cos k_y - t_z^{(1)} \cos k_z\right)\sigma_x + 2t_y^{(2)} \sin k_y \sigma_y + 2t_z^{(2)} \sin k_z \sigma_z. \quad (2)$$

Since the second and third terms vanish at $k_y = 0, \pi$ and $k_z = 0, \pi$ respectively, the gapless points in the energy spectrum along with $k_x$ directions are given by the zeroes of the first term

$$\cos k_x = \left(M - (-1)^\mu\, t_y^{(1)} - (-1)^\nu\, t_z^{(1)}\right)/t_x. \quad (3)$$

Note that here both $\mu$ and $\nu$ take values 0 and 1, depending on the $k_y$ and $k_z$ values being 0 and $\pi$ respectively. The above equation represents a set of four equations whose solutions are the Weyl nodes of the model, each equation potentially producing a pair of Weyl nodes. Since both $\mu$ and $\nu$ take two values each, and each equation can have two values of $k_x$ where the spectrum can be gapless, there could be upto eight distinct Weyl nodes. Of course, not all parameter values support eight solutions. We label the WSM phases with 1, 2, 3 and 4 pairs of Weyl nodes as W2, W4, W6, and W8 in Fig. 1(a) respectively.

We now show how to obtain the phase boundaries in the $(M, t_x)$ parameter space from Eq. 3. The knowledge we gain from here will be useful when we will study the model in a magnetic field in the following sections. The left hand side of Eq. 3 is $\cos k_x$ which lies in the range $[-1, 1]$. Equation 3 has a solution only in the range of parameters for which the right hand side also lies in the same range for a given $\mu, \nu$. Recall that different $\mu, \nu$ correspond to different values of $k_y, k_z$ where the spectrum is gapless. Thus, Weyl nodes at different $\mu, \nu$ cannot annihilate each other. A phase transition corresponds to a change in the number of Weyl nodes at a specific $\mu, \nu$, which can only occur when two Weyl nodes are either created or destroyed. Eq. 3 shows that if $k_{x0}$ is a solution, so is $-k_{x0}$. The two solutions can be created/annihilated only when they coincide in the BZ, which means at $k_{x0} = 0, \pi$. Therefore, at a putative phase transition, the right hand side of Eq. 3 should be at the boundary of its allowed range, namely $\pm 1 = (-1)^\delta$. Thus the condition for the phase transition is,

$$M - (-1)^\delta\, t_x = (-1)^\mu t_y^{(1)} + (-1)^\nu t_z^{(1)}. \quad (4)$$

The above equation represents a set of eight equations because each of the exponent $\delta, \mu$ and $\nu$ take the two values 0 and 1. Clearly, the equations are symmetric under $M \to -M$ and $t_x \to -t_x$, because this does not lead any new conditions - it merely shuffles the set of eight equations. Therefore, we can restrict ourselves to $M, t_x \geq 0$.

In addition to WSM phases and a trivial insulator, we find a layered Chern insulator (LCI) with a bulk gap. The LCI phase can be imagined as a stack (along the $x$-direction) of 2D Chern insulator layers in the $yz$ plane which are tunnel-coupled via $t_x$. When $t_x = 0$, we have a set of disconnected 2D Chern insulator layers for $M < 2$ (for $M > 2$, it is a trivial insulator, see Fig. 1a), with a Chern number $C = 1$ per layer. For $t_x \neq 0$, this simple picture no longer holds, but one can compute the Chern number $C(k_x)$ for each value of $k_x$ by integrating the Berry curvature over the two dimensional $k_y$-$k_z$ BZ. We obtain $C(k_x) = 1$ for each $k_x$ showing that the LCI is adiabatically connected to the $t_x = 0$ limit. The LCI phase has the expected surface state, which is the collection of quantum Hall-like chiral edge states of the 2D Chern insulators, spanning the entire Brillouin zone along $k_x$ direction. Here we note that increasing tunnel-coupling $t_x$ runs a phase transition from LCI to WSM [27] (also clear from the phase diagram Fig. 1a ).

The Fermi arc surface states in all the WSM phases as well as in the LCI phase are (pseudo)spin-polarized along the $z$-direction. The surface states which are localized on the $xy$

crystal surface have the low energy dispersion $E(k_x, k_y) \propto k_y$, and the surface states which are localized on the $xz$ crystal surface have the low energy dispersion $E(k_x, k_z) \propto k_z$. For completeness, we have shown the spectra for some of the phases in Appendix B. We have also shown plots of a few surface states and numerically confirmed their pseudospin polarizations and the directions of their velocities.

# 3 The phase diagrams for $p/q$ flux quanta per unit cell

## 3.1 Hamiltonian in Hofstadter regime

Now we are ready to examine how the phase diagram at zero orbital flux (Fig. 1a) gets modified in the presence of an orbital magnetic field. Note that we consider the orbital effect only. The reason is the following: The orbital coupling, since it couples to the charge degree of freedom, is universal. On the other hand, since the pseudospin label is a **k**-dependent linear combination of spin and orbital labels, which does depend sensitively on the microscopic material parameters, the associated Zeeman coupling is not universal. In any case, the most general form of the Zeeman coupling in our Hilbert space is

$$H_Z = \gamma(\mathbf{k}) B \hat{n}(\mathbf{k}) \cdot \boldsymbol{\sigma} \,. \tag{5}$$

Provided $\gamma(\mathbf{k})$ is not too large, the effect of the Zeeman coupling will be to shift the locations of the Weyl nodes (if present), and shift the phase boundaries. Since the Zeeman coupling is not expected to introduce anything qualitatively different, we will ignore it in the following.

We consider an orbital flux along the $z$-direction perpendicular to the $xy$ plane. The hopping terms in the Hamiltonian pick up a nontrivial phase factor under Peierls substitution [28]. We will work in the Landau gauge $\mathbf{A} = (-y, 0, 0)B$, where only the hopping in the $x$-direction picks up a nontrivial phase so that the Hamiltonian in a magnetic field is obtained from Eq. 1 by the replacement $T_x \to T_x \exp(-i2\pi y\phi/\phi_0)$. We will restrict ourselves to the case where the flux $\phi$ (in units of the quantum flux $\phi_0 = h/e$) per unit cell is commensurate i.e. $\phi = Ba^2/\phi_0 = p/q$, where $p$ and $q$ are relatively prime, so that translation symmetry along the $y$-direction is restored with a larger unit cell [29]. To diagonalize the Hamiltonian, we define a magnetic unit cell which is $q$ times the original unit cell, extended in the $y$-direction. Upon Fourier transformation with respect to the Bravais lattice sites of the magnetic unit cell, the following Bloch Hamiltonian is obtained

$$h_\phi(\mathbf{k}) = \sum_{\alpha=0}^{q-1} c_\alpha^\dagger(\mathbf{k}) \left[ f_1^\alpha(\mathbf{k})\sigma_x + f_3^\alpha(\mathbf{k})\sigma_z \right] c_\alpha(\mathbf{k}) - \left( c_{[\alpha+1]}^\dagger(\mathbf{k}) e^{iqk_y \delta_{(\alpha,q-1)}} T_y c_\alpha(\mathbf{k}) + H.c. \right). \tag{6}$$

We will refer to $h_\phi(\mathbf{k})$ as the Hofstadter Hamiltonian, where $\alpha = 0, 1, ..., q-1$ are the sublattice indices in the magnetic unit cell and **k** lies in the reduced (magnetic) Brillouin zone, *i.e.*, **k**: $k_x \in (0, 2\pi)$, $k_y \in (0, 2\pi/q)$, $k_z \in (0, 2\pi)$. The square bracket notation in the above equation implies that the values are taken modulo $q$ - *i.e.*, $[\alpha] = \alpha \mod q$. The hopping matrix $T_y = t_y^{(1)}\sigma_x + it_y^{(2)}\sigma_y$ has been defined earlier in the previous section. The functions $f_1^\alpha$ and $f_3^\alpha$ are defined as

$$f_1^\alpha(\mathbf{k}) = 2\left( M - t_x \cos\left( k_x + \frac{2\pi p}{q}\alpha \right) - t_z^{(1)} \cos k_z \right), \tag{7a}$$

$$f_3^\alpha(\mathbf{k}) = 2t_z^{(2)} \sin k_z \,. \tag{7b}$$

Note that the spectrum of $h_\phi(\mathbf{k})$, shown explicitly in Appendix C, is particle-hole symmetric. As shown in Appendix C, we can compute the entire phase diagram analytically if we set $|t_y^{(2)}| = |t_y^{(1)}|$. To avail ourselves of the simplicity and computational advantages this gives us, especially at large $q$, we will make this choice $t_y^{(2)} = t_y^{(1)} \equiv t_y$ for the rest of the paper.

## 3.2 Gapless points and phase boundaries

To obtain the phases and phase boundaries, we need to identify the zeroes of the Hofstader Hamiltonian $h_\phi(\mathbf{k})$ (its spectrum is particle-hole symmetric about the zero energy as shown in Appendix A), which gives us the band-touching points where the energy spectrum vanishes, which in turn allows us to find the number of Weyl nodes. This can be done by explicitly writing the Bloch Hamiltonian as a $2q \times 2q$ matrix in the basis of sublattice and pseudospin $\Psi = \left( c_{0,\uparrow}, c_{1,\uparrow}, ..., c_{q-1,\uparrow}, c_{0,\downarrow}, c_{1,\downarrow}, ..., c_{q-1,\downarrow} \right)$. The details of the calculation are shown in Appendix C. The energy gap can close only at $k_y = 0$ and/or $k_y = \pi/q$, and only at $k_z = 0$ and/or $k_z = \pi$. For each of these possibilities, the $k_x$ values at which the spectrum is gapless are given by

$$\cos q k_x = (-1)^{p+q} \left[ -T_q(g) + (-1)^{\mu-q} \, 2^{q-1} \, (t_x/t_y)^{-q} \right], \tag{8}$$

where $\mu$ takes values 0 and 1 which correspond to closing of the gap at $k_y = 0$ and $k_y = \pi/q$ respectively. Here $T_q(g)$ is the Chebyshev polynomial [30] of the first kind of degree $q$. Its argument $g = \left( -M + (-1)^\nu t_z^{(1)} \right)/t_x$, where $\nu$ takes values 0 and 1, which correspond to closing of the gap at $k_z = 0$ and $k_z = \pi$ respectively. So essentially, Eq. 8 is a set of four independent equations. Note also that Eq. 8 involves only the parameters $g$ and $t_x/t_y$. Finally, we note that for a given $(g, t_x/t_y)$, if $k_x = k_0$ is a solution of Eq. 8, then so is $k_x = -k_0$. Furthermore, since $\cos q(\pm k_0 + 2\pi m/q) = \cos q k_0$, it is easy to see that $k_x = k_0 + 2\pi m/q$ where $m = 0, 1, 2, ..., (q-1)$ are also solutions. Since, without loss of generality, $k_0$ can be restricted to $0 \le k_0 \le 2\pi/q$, it is clear that there are $2q$ values of $k_x$ in the magnetic BZ where the spectrum could be gapless. Since $\mu$ and $\nu$ take two values each, and each equation can have $2q$ values of $k_x$ where the spectrum can be gapless, there could be regions in the parameter space with up to $8q$ distinct gapless values for $k_x$. These distinct gapless points in the magnetic BZ are the Weyl nodes in the theory.

We can now obtain the phase boundaries from Eq. 8 by arguments identical to those given in the zero field case in Sec. 2 above Eq. 4. As the left hand side is now changed to $\cos q k_x$ the number of zeros will be $2q$ for a given $\mu, \nu$ and the phase boundary equation will be (after setting RHS to $\pm 1$ in Eq. 8),

$$-T_q(g) + (-1)^{\mu-q} \, 2^{q-1} \, (t_x/t_y)^{-q} = (-1)^{p+q+\delta}, \tag{9}$$

where $\delta$ takes two values 0 and 1, which correspond to setting the RHS of Eq. 8 equal to 1 and $-1$ respectively. Analogous to Eq. 4 in the zero-field case, the above compactly written equation is a set of eight independent equations because each of the exponents $\mu$, $\nu$ and $\delta$ take two values. Note that the only appearance of the numerator $p$ (of the flux $p/q$) is in the exponent on the RHS, and the value only depends on whether $p$ is odd or even. This only shuffles between the set of equations and does not lead to any new condition. So the phase boundaries are completely independent of the value of $p$. Interestingly, as we mentioned earlier, there are only two parameters $g$ and $t_x/t_y$ which enter the Eq. 9. The hopping parameter $t_z^{(1)}$ enters the equation via the value of $g = \left( -M + (-1)^\nu t_z^{(1)} \right)/t_x$, which, given that a solution exists, determines the values of $k_x$ at which there are Weyl touchings. To simplify our analysis without sacrificing anything qualitatively new, we henceforth fix $t_z^{(1)} = t_y^{(1)} = t_y^{(2)} = t_y = 1$

and $t_z^{(2)} = 1$. With this simplification, our phase diagram is essentially controlled only by two parameters $M/t_y$ and $t_x/t_y$. The same Eqs. 8-9 are also applicable to the zero field case and correctly reproduce (setting $q = 1$ in Eq. 9) the phase boundaries given by Eq. 4.

Note that a uniform magnetic field in the $y$-direction, would give identical conditions (Eqs. 8 and 9) for the gapless spectrum and the phase boundary respectively, if we make the replacements $t_z^{(1)} \leftrightarrow t_y^{(1)}$, $t_z^{(2)} \to -t_y^{(2)}$ and $t_y^{(2)} \to t_z^{(2)}$. So all our results including the phase diagrams described in the following sections apply to this case as well.

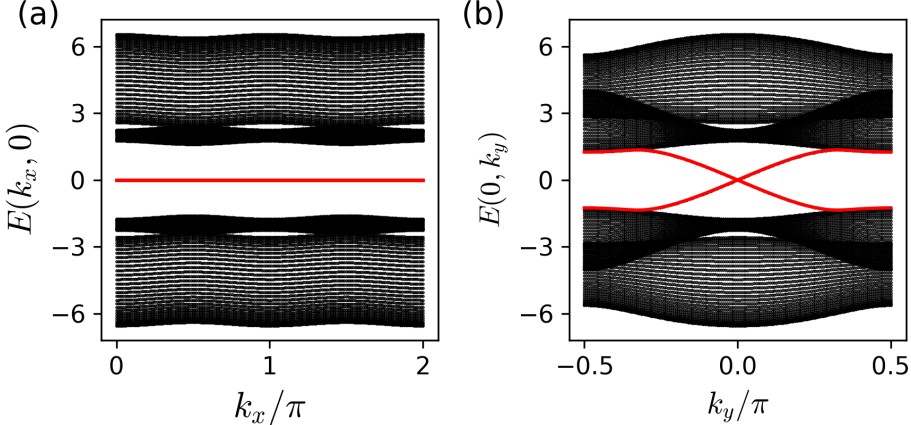

Figure 2: The spectrum of a finite slab in the $z$-direction in the LCI phase, when $q = 2$. The parameters are $M = 1.2t_y$, $t_x = 0.4t_y$. (a) The surface states (highlighted in red) are dispersionless in $k_x$ and span the BZ. (b) The surface states (highlighted in red) disperse in $k_y$, with linear dispersion near zero energy, with two gapless modes of opposite slopes belonging to opposite open surfaces of the slab. The surface state living on the $xz$ crystal surface, has a linear dispersion in $k_z$ near zero energy (not shown here).

### 3.3 Phase diagrams for a strong field, small q

In the earlier section, we obtained an analytical expression for the phase boundaries at any arbitrary commensurate magnetic flux penetrating a primordial unit cell $\phi = Ba^2/\phi_0 = p/q$, where $p$ and $q$ are relatively prime. As explained earlier, the phase boundaries are independent of $p$, and the phase diagrams are shown in Fig. 1 for $q = 1, 2, 3$. Note that $q = 1$ implies that an integer quantum of flux pierces the primordial unit cell. Since there are no nontrivial phase factors in the hopping terms, we get back the zero-field Hamiltonian for $q = 1$. From Fig. 1, we see that the phase diagrams for $q = 2, 3$ are similar to the diagram for $q = 1 \Rightarrow B = 0$. However, there are certain prominent differences -

(i) Instead of the gapless phases with 2, 4, 6, and 8 Weyl nodes, we now have gapless phases with 2q, 4q, 6q, and 8q Weyl nodes. This $q$-fold increase in the number of Weyl nodes is a consequence of the translation symmetry of the Hamiltonian in the $y$-direction within a magnetic unit cell.

(ii) For non-zero magnetic fields, there are two additional phases which we call W2′ and $I'$, which do not occur for $B = 0$. The spectrum of the phase W2′ has Weyl nodes in certain bands, while other bands form an LCI. Surface states of both types are present. The phase $I'$ is a trivial insulator in the bulk but has counter-propagating surface states on certain surfaces.

(iii) Each of the WSM phases gets fragmented and multiple copies appear with increasing $q$. We can understand this as follows. Each of the critical curves (phase boundaries) are determined by Eq. 9, which are inherently $q^{th}$ degree polynomials in $(M, t_x)$. Thus there are many solutions with increasing $q$.

(iv) Despite its increasing complexity with increasing $q$, the phase diagram approaches a limit in the $q \to \infty$ limit, with only four nontrivial phases W4, W8, LCI, and $I'$ (see Fig. 8). We will describe in detail the weak-field limit behavior in Sec. 5.

(v) The gapped LCI and $I'$ phases take up regions from the WSM phases and expand with increasing $q$ and approach a limit when $q$ is large (see Fig. 8). Therefore a strong magnetic field can drive a Weyl semimetal to an insulator with a robust gap ($\sim t_y$, when $p/q \sim 1/2 \Rightarrow B \sim 10^3$ Tesla, for lattice constant $a \sim 1$ nm) in our lattice model (see Sec. 5).

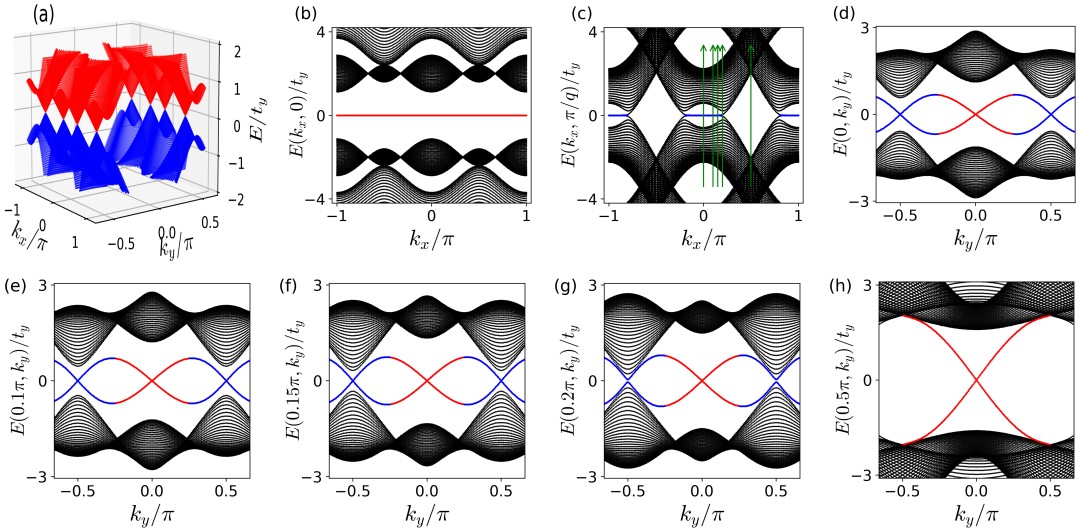

Figure 3: Bulk and surface spectrum in the W2$'$ phase for $q = 2$, for $M = 1.2t_y$ and $t_x = 1.3t_y$. (a) The bulk bands around zero energy show $2q = 4$ Weyl nodes which live at the boundary of the magnetic BZ ($k_y = \pm\pi/q$, $k_z = 0$). In (b)-(h) the spectrum along $k_x$ and $k_y$ directions is shown for the WSM slab (finite in the $z$-direction with $L_z = 40$). (b)Spectrum along $k_x$ for a fixed $k_y = 0$ shows the surface states (in red) which are not connected to the bulk Weyl node projections and span the magnetic BZ. (c) Spectrum along $k_x$ for a fixed $k_y = \pi/q$ shows the Fermi arc surface states (in blue) connecting the bulk Weyl node projections. (d)-(h) Spectrum along $k_y$ for a series of values of $k_x$ (indicated in (c) by vertical green arrows) shows the existence of two types of surface states. The surface states around $k_y = \pm\pi/q$ correspond to the Fermi arc surface states which are connected to the bulk Weyl node projections, whereas those around $k_y = 0$ are the additional surface states unconnected to the bulk. The red and blue shadings are meant for visualizations of the surface states around $k_y = 0$ and $k_y = 0.5\pi$ respectively. As the Weyl point projection is approached, the associated Fermi arc surface states start to mix with the bulk states (see figure (g)) and the decay length will diverge when we hit the Weyl node. In (h) we have only one type of surface states. The decay length of the surface states (shown in Figs. 3e-h) for a series of $k_y$ values in between $k_y = 0$ and $0.5\pi$ is plotted in Fig. 4.

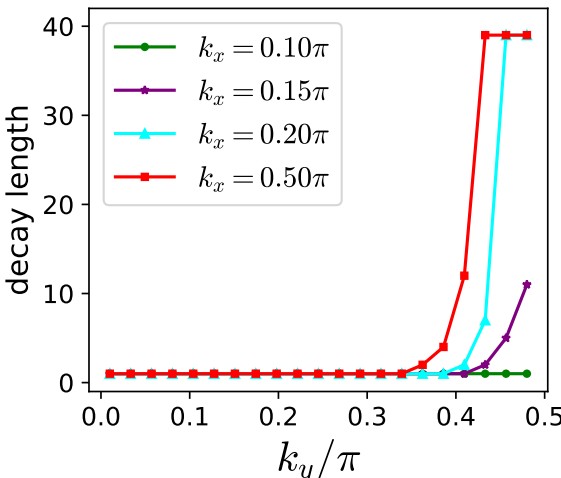

Figure 4: Decay length plot of the surface states shown in Figs. 3e-h, for a series of $k_y$ values in between $k_y = 0$ and $0.5\pi$. Surface states at around $k_y = 0$ do not decay into the bulk for any $k_x$ values, but those at around $k_y = 0.5\pi$ start decaying into the bulk when $k_x$ is taken closer to the Weyl point projection $k_x \approx 0.21\pi$, $k_y = 0.5\pi$. When $k_x = 0.20\pi$ is very close to Weyl point projection, decay length of the surface states at around $k_y = 0.5\pi$ diverges (hits the system size). At $k_x = 0.5\pi$, we have only one type of surface states (but no Fermi arc states associated with Weyl nodes) which mixes with bulk for $k_y$ values far away from $k_y = 0$ (which can be clearly seen in Fig. 3h). We compute decay length $l$ by solving $\ln\left[|\psi(z=0)|^2\right] = 1 + \ln\left[|\psi(z=l)|^2\right]$ numerically, where $\psi(z)$ is the wavefunction of the surface states.

### 3.3.1 The LCI phase

As explained earlier, the LCI phase can be thought of as a stack (along x-direction) of 2D Chern layers which are tunnel coupled via the hopping $t_x$. The phase is adiabatically connected to the set of disconnected Chern layers in the limit $t_x \to 0$. The presence of an external magnetic field which is parallel to the Chern layers should not change the LCI phase. In fact, we find that the LCI phase is always present in the phase diagram for all $q$. We also note that some parts of the WSM phases in the phase diagram of Fig. 1a transform to the LCI phase in the presence of the magnetic field.

Other than the LCI and $I'$ phases, all other phases get fragmented into multiple copies as $q$ is increased. For example, there are $2, 3, 5, 5$, and $3$ copies of W2, W2′, W4, W6, and W8 respectively for $q = 3$. The LCI phase, which exists for arbitrary $q$, expands both along $M$ and $t_x$ directions with increasing $q$ (see Fig. 1) by making the system gapped in a larger region of the parameter space. It is instructive to find the maximum values of $M$ and $t_x$ upto which the region occupied by the LCI can reach for a given $q$ and also determine the limits of these maximum values as $q \to \infty$. So we define the critical values $t_x^c$ and $M^c$ as follows: for $t_x > t_x^c$ the LCI phase does not exist for any $M$, and similarly for $M > M^c$ the LCI phase does not exist for any $t_x$ for a given $q$. For a given $q > 1$, it is possible to obtain the exact expressions for these critical values (details in Appendix D),

$$t_x^c = t_y \, 2^{1-1/q} \,, \tag{10a}$$

$$M^c = t_z^{(1)} + t_x^c \cos(\pi/2q) = \left(1 + 2^{1-1/q} \cos(\pi/2q)\right) t_y \,, \tag{10b}$$

where we set $t_z^{(1)} = t_y$ in the second equation. For $q = 1$, we have $t_x^c = t_y$ and $M^c = 2t_y$. As

$q \to \infty$, these critical values approach $t_x^c = 2t_y$ and $M^c = 3t_y$ (see the large $q$ phase diagram in Sec. 5). So we find that the gapped LCI phase is always confined in a finite region $M < M^c$, $t_x < t_x^c$ for all values of magnetic field strength.

### 3.3.2 WSM phases and Chern numbers

Let us now examine the WSM phases in more detail. We know that in the zero-field WSM phase, it is possible to understand the existence of the surface Fermi arc states by computing the Chern numbers [2] of the two-dimensional planes that cut the Fermi arc. A similar analysis here is slightly more subtle. Since in our model, the Weyl nodes are all along the $k_x$ axis, on the $k_x$-$k_y$ surface BZ, the projection of all the Weyl nodes are on the lines $k_y = 0$ and/or $k_y = \pi/q$. In general, there are two sets of Fermi arc surface states on the surface BZ. One set is at $k_y = 0$ and the other is at $k_y = \pi/q$ (see Fig. 6 for an illustration). In both cases, the Fermi arcs are dispersionless along the $k_x$-direction. We can obtain the total Chern number $C(k_x)$ by integrating the Berry curvature of all the occupied bands of the two-dimensional magnetic $k_y$-$k_z$ BZ. In our chosen Landau gauge $\mathbf{A} = (-y, 0, 0)B$, as earlier discussed in the beginning of Sec. III, we get $C(k_x) = 1$ (or $-1$), when $k_x$ belong to the Fermi arc states at $k_y = 0$ (or $k_y = \pi/q$) respectively for all the planes that cut a single Fermi arc. If the constant $k_x$ plane cuts both the Fermi arcs at $k_y = 0$ and $k_y = \pi/q$, then we get $C(k_x) = 0$. However, a computation of the same quantity in a different Landau gauge $\mathbf{A} = B(0, x, 0)$ gives $C(k_x) = q, -q$ and $0$ for each of the three cases above. The reason is that what is physical, and thus gauge-invariant, is the Chern number per unit length in the $x$-direction (directly proportional to the Hall current per unit $x$-length). In the first Landau gauge, the magnetic unit cell is the same size in the $x$-direction as the primordial unit cell, while in the second Landau gauge the magnetic unit cell is $q$-fold longer in the $x$-direction than the primordial one. The Chern number per unit length in $x$ is the same in both cases.

### 3.3.3 The phase W2′

Now let us turn to the W2′ phase which did not exist when $B = 0$. This is a gapless phase with $2q$ Weyl nodes. Its novelty lies in the fact that it has two types of bulk bands: those that touch at Weyl nodes, and those that are fully gapped but carry a nonzero Chern number. Thus, the W2′ is a phase shows the coexistence of WSM and LCI bands. Consequently, it has two types of localized surface states - (i) the standard Fermi arc surface states which are connected to the bulk states at the surface projection of the Weyl nodes and (ii) surface states which are disconnected from the Weyl nodes, and span the magnetic BZ (see Fig. 3). The most important difference between the two types of surface states is their decay length into the bulk. The decay lengths of the Fermi arc surface states diverge at the surface projection of the Weyl nodes whereas the decay length of the surface states associated with the LCI bands always remains finite. In this respect, they resemble the surface states of a topological insulator. The two types of surface states are also depicted in mixed real and momentum space representation in Figs. 5a-5b.

The Hall response of the W2′ phase is additive between the two types of bands. The Weyl semimetal Fermi arc surface states contribute $k_0(q)qe^2/2\pi h$ to the Hall conductance per layer [27], while the layer Chern insulator surface states will contribute $e^2/h$ to the Hall conductance per layer. Since their surface states are counter propagating, the total effective Hall conductance per layer is

$$\sigma_{yz} = \frac{e^2}{h} - \frac{e^2}{2\pi h} q k_0(q)$$
$$= \frac{e^2}{h} \left(1 - \frac{q k_0(q)}{2\pi}\right), \tag{11}$$

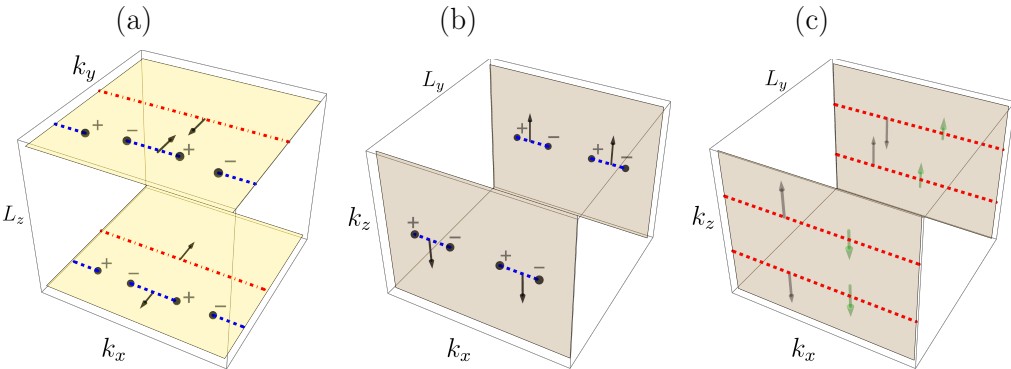

Figure 5: The zero-energy Fermi arc surface states shown in Fig. 3 in the W2$'$ phase, are depicted here through the cartoon figures (a) and (b), in mixed real and momentum space representation. (a) In the $k_x$-$k_y$ surface BZ, there are two types of surface states: type (i) surface states (shown in blue), which are the standard Fermi arcs, end at the Weyl point projections ; type (ii) surface states (shown in red) live across the surface BZ. The velocity $\mathbf{v} = \nabla E(k_x, k_y)$, of the particles occupying the type (i) surface states are along the positive/negative $y$-direction on the top/bottom surface. For type (ii) surface states, the direction of the velocity gets reversed. So type (i) and type (ii) surface states are counter propagating. The Chern number $C(k_x) = 1$, if the constant $k_x$ plane cuts only the type (ii) surface states, else it is zero. (b) In the $k_x$-$k_z$ surface BZ, there are only type (i) Fermi arc surface states which end at the Weyl point projections. Here we note that the W2$'$ phases which share a boundary with the $I'$ phase (see Fig. 1c) have two types of surface states which co-exist on the $k_x$-$k_z$ surface BZ. (c) This cartoon figure shows the zero-energy surface states in the $I'$ phase, in mixed real and momentum space representation. In the $I'$ phase, there exists localized surface states only on the $k_x$-$k_z$ surface BZ. On both the open crystal surfaces $y = 0$ and $y = L_y$, there are two counter propagating surface states with green arrows indicating their spin polarization. Here, the Chern number $C(k_x) = 0$, for all $k_x$ planes.

where $k_0(q)$ is the length of each of the $q$ Fermi arcs connecting the $2q$ number of Weyl point projections on the $k_x$-$k_y$ surface BZ where both the LCI and Weyl semmimetal Fermi arc surface states exist together (see also Fig. 5a). We also note that the length of the Fermi arc $k_0(q)$ changes (it decreases) with increasing $q$. So generally the Hall conductance in the W2$'$ phase can be tuned by changing the external applied magnetic field.

A similiar phenomenon involving the co-existence of a Weyl semimetal band and a gapped topological phase has also been recently studied in Ref. 31. However, in their case, the WSM has higher order Weyl nodes, and the topological bands are part of a higher order topological phase with hinge states which require crystalline symmetry. In our case, we have a simple Weyl semimetal and a LCI which co-exist, with no crystalline symmetry needed.

### 3.3.4 The phase $I'$

The phase $I'$ is a new topologically trivial phase that appears in the presence of a magnetic field and is insulating in the bulk. However, it has zero energy surface states which exist only on the $k_x$-$k_z$ surface BZ (010 crystal surface). Each of the open crystal surfaces $y = 0$ and $y = L_y$ host a pair of counter-propagating states which are separated in $k_z$ (see Fig. 5c). A physical way to think about this insulator is as having bulk gapped bands carrying opposite Chern numbers. The projections of the gapless surface states corresponding to these bulk

bands are naturally counterpropagating, and separated in $k_z$. If the free surface of the slab is parallel to the $xy$-plane, $k_z$ is not a good quantum number, and the would-be surface states hybridize and gap themselves out. However, if the free surface is parallel to the $xz$-plane then $k_z$ is a good quantum number, and they cannot gap themselves out.

To support this picture, note that $I'$ always appears adjacent to W2'. From W2', one can obtain $I'$ by the expanding the separation of the Weyl nodes till the Fermi arc stretches across the BZ before allowing them to annihilate (see Fig. 5 and caption therein).

# 4 Bulk and surface dispersions

In this section, our main aim is to understand how the bulk and the surface band structures evolve as we vary the parameters to approach the phase boundaries. We first study the bulk and surface bands deep inside each phase and then analyze how the phase transitions that occur via the creation/annihilation of Weyl nodes alter the bulk and surface dispersions.

## 4.1 Bulk dispersion within the phase

The spectrum of the $2q \times 2q$ Hamiltonian given in Eq. 6 has been explicitly derived in Appendix C:

$$E_{n,r}(\mathbf{k}) = r\sqrt{\gamma_n(\mathbf{k}) + (2t_z^{(2)}\sin k_z)^2}\,, \tag{12}$$

with $n = 1, 2, 3, ..., q$ and $r = (-, +)$. Although we do not have a closed form expression for $\gamma_n(\mathbf{k})$ (which is non negative for all $\mathbf{k}$) we can numerically compute its dependence on the parameters $(M, t_x, t_y, t_z^{(1)})$ for the flux $p/q$. We note that in all the gapless Weyl semimetal phases including the W2' phase, the low energy bulk dispersion around each of the Weyl nodes $\mathbf{k}_0$, is linear in all the three directions $k_x, k_y$ and $k_z$, for all values of $q$. This will change close to the phase boundaries. In the gapped phases, which are the LCI, $I'$ and the normal insulator (NI), the spectrum has a gap at the Fermi energy (zero-energy).

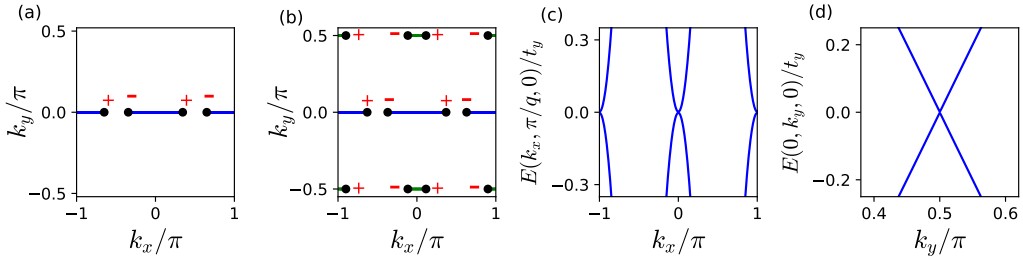

Figure 6: Fermi arcs on the surface BZ $k_x$-$k_y$ in (a) W2 and (b) W4 for $q = 2$. $M$ is taken to be $2.2t_y$, and $t_x$ is tuned from $t_x = 1.4t_y$ in W2 to $t_x = 1.65t_y$ in W4, which gives rise to a new pair of Weyl nodes around $k_x = 0$ and $\pm\pi$, at $k_y = \pm\pi/q$, $k_z = 0$. The low energy bulk dispersions exactly at the phase boundary $M_c = 2.2t_y$, $t_x^c = 1.56t_y$, are shown in (c) and (d). The Weyl nodes get created/annihilated via a band touching which is (c) quadratic in $k_x$ and (d) linear in $k_y$. The dispersion along $k_z$ is also linear (not shown here).

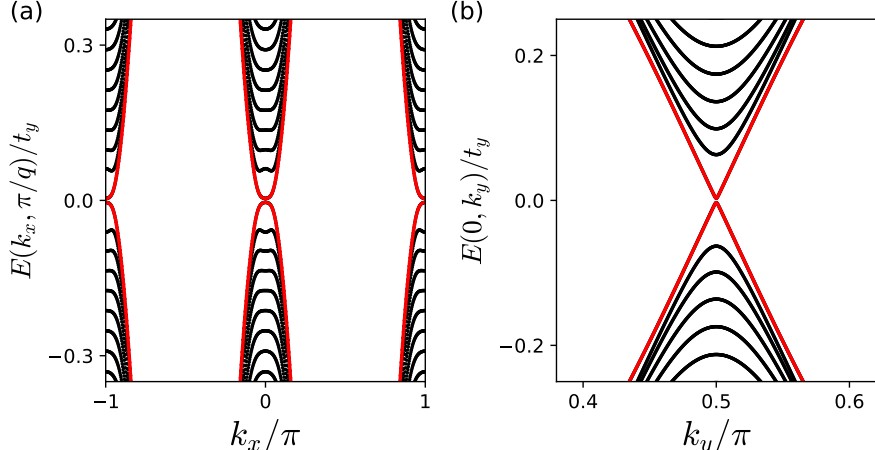

Figure 7: The low energy dispersion of the surface states when the system is a finite slab along the $z$-direction with $L_z = 80$, for $q = 2$. The parameters $M$ and $t_x$ are chosen so that the system is at the boundary between phases W2 and W4. See the caption of Fig. 6 for the choice of parameters. The surface bands (red) touch (a) quadratically in $k_x$ and (b) linearly along $k_y$. The low energy surface band touching is similar to the low energy bulk band touching (see Fig. 6c-d). The small gap at zero energy in the spectrum is due to finite size effects.

### 4.2 Bulk dispersions at phase boundaries

The boundaries between the phases in the phase diagram (see for example, Figs. 1b-c), can be between (a) gapped and gapless phases (b) two gapless phases, and (c) two gapped phases. Category (c) never occurs in our model, as a gapless phase always intervenes between two gapped phases, even if its area in parameter space is exponentially very small. For instance in Fig. 8, the intervening gapless phases are not visible in the phase diagram. The absence of a direct transition between two gapped phases is a consequence of the fact that the band structure is continuous in the parameters of our Hamiltonian. For both categories (a) and (b), the transitions occur through a band touching which is quadratic in $k_x$, but linear in $k_y$ and $k_z$. At these transitions, pairs of Weyl nodes are either created or annihilated. An illustrative case is depicted in Fig. 6.

### 4.3 The dispersion of the surface states within the phases and at boundaries

For WSM phases, Fermi arc states are obtained by diagonalising the Hamiltonian of a finite slab along either the $z$ (or $y$) directions. In the surface BZ $k_x$-$k_y$ (or $k_x$-$k_z$), they are dispersionless along $k_x$ and linear in $k_y$ (or $k_z$). The relevant dispersions in the W2 and W4 phases are shown in Fig. 6. More details on the dispersion of the surface states in the W2′ phase are shown in Fig. 3 and for the $I'$ phase in Fig. 5c.

As in the bulk spectrum, at phase boundaries, the surface bands touch quadratically in $k_x$, and linearly in the remaining direction $k_y$ (or $k_z$) when the system is taken to be finite along the z (or y) direction. An example is shown in Fig. 7. One might think that interactions could play an important role on the 2d surface of the sample because the combination of the quadratic dispersion in $k_x$ and the linear in the other direction could lead to a finite density of states (DOS) at the Fermi energy. However, it turns out that the low energy DOS goes as $\sqrt{E}$, and vanishes at zero energy.

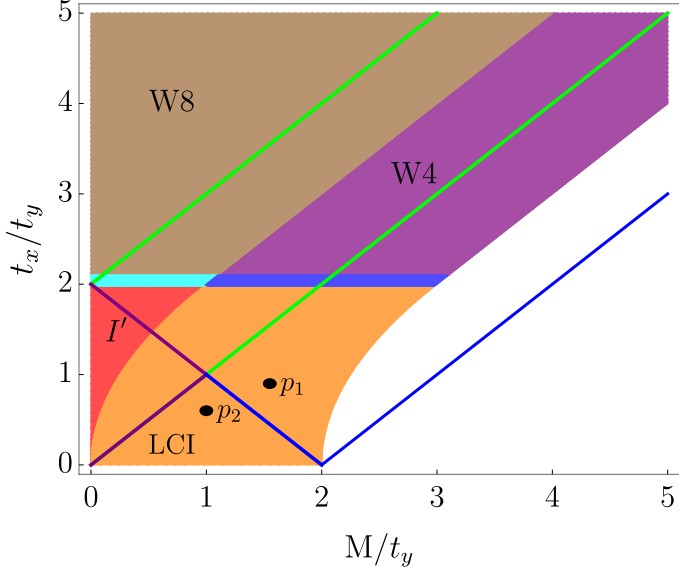

Figure 8: The phase diagram for $q = 200$. The blue, green and purple straight lines are the phase boundaries when $B = 0$ (see Fig. 1a for details). Clearly, for very large $q$, the phase diagram looks much simpler with just four nontrivial phases W8 (brown), W4 (purple), LCI (orange) and $I'$ (red), and of course the trivial insulator phase. We note that the regions occupied by $I'$, a significant part of the LCI and the normal insulator (NI) above the blue straight line were semimetallic when the external magnetic field was zero. As $q \to \infty$, the widths of the blue region which consists of many copies of W2 and W2$'$ phases as well as the cyan region which consists of many copies of W2, W2$'$, W4, W6, and W8, shrink to zero at $t_x^c = 2t_y$. Along the $M$ direction, the LCI phase expands upto $M^c = 3t_y$ in the $q \to \infty$ limit. The critical values $M^c$ and $t_x^c$ are derived in Appendix D and also discussed in Sec. 3.3.1. The points $p_1$ and $p_2$ are discussed in the text.

## 5 The Weak-Field Limit

So far we have studied the phase diagram for small values of $q$. In this section, we will focus on the phase diagram in the weak magnetic field limit where $B \to 0$. The $B = 0$ or zero flux case, corresponding to $q = 1$, has been studied earlier in Sec. 3. However, the $q \to \infty$ limit, where the magnetic flux $\phi \propto 1/q$ goes to zero, is more subtle. Equations 8 and 9, do not have any simple behaviour as $q \to \infty$. Nevertheless, the Eqs. 8-9 allow us to construct the phase diagram for large values of $q$ and consequently deduce the phase diagram in the limit $q \to \infty$. Surprisingly, the phase diagram has a simple structure in this limit, with only four nontrivial phases occupying a significant part of the parameter space - W8, W4, LCI and $I'$. This is shown in Fig. 8. We have checked that the widths of the blue region which consists of multiple copies of the W2 and W2$'$ phases and the cyan region which consists of multiple copies of the W2, W2$'$, W4, W6, and W8, shrink to zero when $q \to \infty$. Similarly, a thin sliver of the W2$'$ phase lying between the LCI and $I'$ phases shrinks to zero as $q \to \infty$. Clearly, this phase diagram looks different from the zero-field phase diagram shown in Fig. 1a (phase boundaries at $B = 0$ have been drawn as lines in Fig. 8 to facilitate the comparison).In particular, regions which are gapless at $B = 0$ appear to be gapped in Fig. 8. On physical grounds, we expect the behavior as $B \to 0$ to be smoothly connected to that at $B = 0$.

To resolve this apparent contradiction, we carry out a more detailed study of the bulk energy gaps and the low energy dispersions at different points in the phase diagram. In what

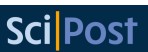

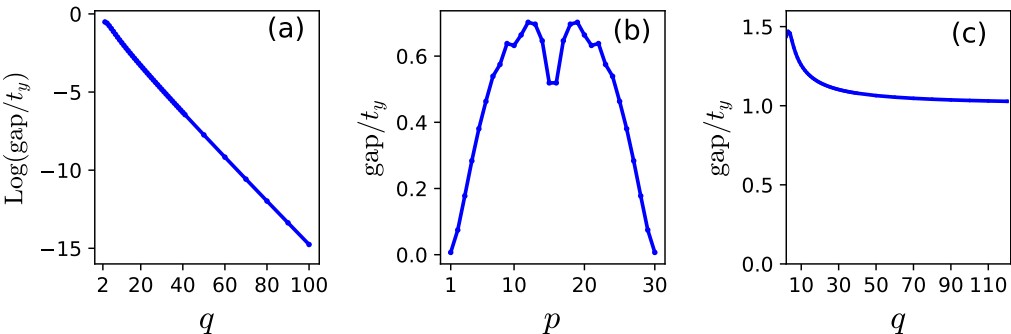

Figure 9: (a) The logarithm of the bulk energy gap (computed numerically), corresponding to a WSM region at $B = 0$, plotted as function of $q$, for a fixed value of $p = 1$, shows the exponential fall as $e^{-\eta_1 q}$, with a constant $\eta_1$. The parameters are $M = 1.8 t_y$, $t_x = 0.9 t_y$ correspond to the point $p_1$ in Fig. 8. (b) The bulk energy gap is plotted as a function of $p$ for a fixed value of $q = 31$. The energy gap first increases with p and then falls again symmetrically about $p \sim q/2$. (c) Bulk energy gap plotted as function of $q$, for a fixed $p = 1$, for the point $p_2$ in Fig. 8, corresponding to a LCI region at $B = 0$. The parameters are $M = 1.2 t_y$, $t_x = 0.4 t_y$. The gap can be fit to a phenomenological form $\Delta(q) \simeq \Delta_\infty e^{q_0/q}$, which does not vanish as $q \to \infty$.

follows, we will fix $p = 1$, unless stated otherwise, and examine large values of $q$, choosing a few representative points in the phase diagram in Fig. 8 to show how the physics as $B \to 0$ is smoothly connected to that at $B = 0$.

Let us consider an arbitrary point $p_1$ in a region which is a WSM when $B = 0$, but is in the LCI phase when $B \to 0$, or at least when $q$ is large. To check whether the phase is really semi-metallic or insulating when $q \to \infty$, we need to check whether the gap vanishes or remains finite in that limit. To study this, we compute the bulk energy band gap at $p_1$ as a function of $q$, shown in Fig. 9a. It can be seen that the gap at zero energy falls exponentially with $q$ at large $q$ for fixed $p = 1$.

To enable the reader to visualize the change in the spectrum graphically, the bulk energy spectrum at the point $p_1$ in the phase diagram is shown in Fig. 10 for two different values, $q = 3$ and $q = 30$ (the magnetic field is along the $z$ direction as earlier). The dispersions along the $k_x$, $k_y$ and $k_z$ directions have been shown explicitly for both cases. The dispersion in the $k_x$ and $k_y$ directions become almost flat for $q = 30$, corresponding to the semiclassical Landau levels. Furthermore, we note that already for $q = 30$, the two central bands come close together at $k_z = 0$ with a very tiny gap, approaching the linear dispersion, in the field direction of the $n = 0$ Landau levels in the semiclassical limit. In fact, even for $q = 30$, the low energy dispersion for small $k_z$ in our lattice model looks almost identical to the dispersion of Weyl fermions in the continuum model [12, 15, 32, 33]. In the $q \to \infty$ limit, clearly, this is the W2 phase (there are only two linearly dispersing $n = 0$ Landau levels). This is true even though, for any finite $q$, the phase is gapped and is in the LCI phase. It is just that the gap is exponentially small in $q$. We have also checked that when the point $p_1$ is moved to regions where the zero field case is in the W4 or W6 phase, the bands come close together both at $k_z = 0$ and $k_z = \pi$, signifying that in the $q \to \infty$ limit, they evolve to Weyl nodes at both $k_z = 0, \pi$.

Now let us consider another point $p_2$ in a region which is a LCI both when $B = 0$ and $B \to 0$. Here, when we compute the bulk energy gap as a function of $q$, we find that it behaves roughly as $\Delta \simeq \Delta_\infty e^{q_0/q}$, which does not vanish as $q \to \infty$. This is shown in Fig. 9c. Further, we also note that in regions which are gapless semi-metals, both when $B = 0$ and $B \to 0$,

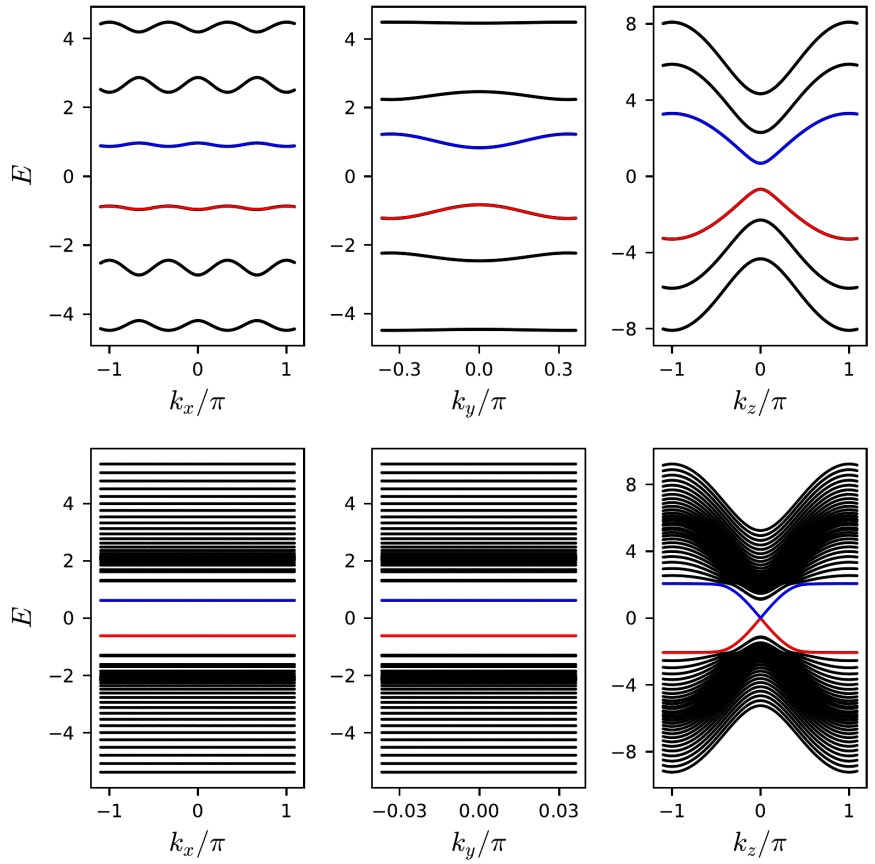

**Figure 10:** The bulk energy spectrum in $k_x$, $k_y$ and $k_z$ directions for $q = 3$ (first row) and $q = 30$ (second row). The parameters are $M = 1.8$, $t_x = 0.9$, at the location of the point $p_1$ in the phase diagram Fig. 8. For each of the plots, one of the momenta is allowed to vary, and the other ones are fixed at $k_x = 0.3\pi$, $k_y = 0.2\pi/q$ and $k_z = 0.1\pi$ appropriately. The two central bands are shown in color. Note that the dispersions in $k_x$ and $k_y$ become almost flat for $q = 30$. Also, the low energy dispersion as a function of $k_z$ for $q = 30$, has a very tiny gap, explained further in the text.

the low-energy dispersion of the $B \to 0$ phase approaches that of the $B = 0$ phase. We hence conclude that although naively the $q \to \infty$ phase diagram looks different from the zero-field ($q = 1$) phase diagram, the physics they describe, in terms of their bulk energy gaps and their low-energy dispersions, is smoothly connected via the semiclassical description.

We end by noting the behaviour of the bulk energy gap as a function of $p$ and $q$, which is shown in Fig. 9. The energy gap decreases with $q$ for a fixed $p$ and increases with $p$ for a fixed $q$, and the maximum gap $\sim t_y$ occurs when $p/q \sim 1/2$. Thus for large $q$ values and when $p/q \sim 1/2$, a very strong magnetic field $B = \frac{p}{q}\frac{\phi_0}{a^2} \sim 10^3$ Tesla (if lattice constant $a = 1$ nm) can drive a Weyl-semimetal to an insulator by annhilating the Weyl nodes with gap $\sim t_y$ in a robust region of the parameter space.At a large magnetic field, annihilation of Weyl nodes is expected to open a gap when the inverse magnetic length $\sqrt{eB/\hbar} \sim k_0$, the separation between the Weyl nodes. This has been observed also in experiments by measuring the resistivity of Weyl materials TaP [34] and TaAs [35] in high applied magnetic field.

# 6  Summary and Outlook

In summary, we have studied a time-reversal broken WSM in the presence of a commensurate orbital magnetic field with minimal crystalline symmetry. We have considered the case where the direction of the magnetic field is normal to the line joining the Weyl nodes in the absence of a magnetic field, which we denote as the $x$-direction. In the presence of a $p/q$ flux per unit cell, we have obtained the phase diagram in the parameter space of the onsite mass $M$ and the $x$-direction hopping $t_x$. Setting two of the five independent hoppings equal ($|t_y^{(1)}| = |t_y^{(2)}|$) allows us to solve for the entire phase diagram analytically for arbitrary $p, q$.

We find that the phase diagram contains WSM phases hosting $2q$, $4q$, $6q$, and $8q$ Weyl nodes. These phases occur in multiple copies for nonzero flux, with the number of copies depending on $q$. The gapped LCI phase also exists for arbitrary $q$. There are two additional phases, which we call W2$'$ and $I'$, that appear in the phase diagram only at nonzero flux. The phase W2$'$ has a gapless bulk spectrum with $2q$ Weyl nodes, but has additional gapped bulk bands which carry nontrivial Chern number in the $k_y$-$k_z$ plane at fixed $k_x$. This phase displays a coexistence of Weyl semimetal and layered Chern insulating behavior. In accordance with the bulk-boundary correspondence, the W2$'$ phase has Fermi arc states at the surface, as well as surface states required by the layered Chern insulator. These two types of surface states in the W2$'$ phase are counter-propagating. The phase $I'$ is fully gapped in the bulk but hosts a pair of counter-propagating surface states on the $xz$ crystal surface, but none on the $yz$ surface.

The fact that we can analytically obtain the entire phase diagram for arbitrary $p, q$ enables us to systematically study the weak-field limit $q \to \infty$, and to smoothly relate it to the $B = 0$ limit. Formally, for $p = 1$ and any large but finite $q$, the phase diagram looks quite different from the zero-field case $p = q = 1$. However, an examination of the spectral gap at zero energy reveals the way the limit should be taken physically. In regions of parameter space where the zero-field case is fully gapped, the $q \to \infty$ gap remains finite when the limit is taken, whereas in the regions of parameter space where the $q = 1$ case is gapless and the large $q$ case appears gapped, the gap of the latter vanishes exponentially as a function of $q$.

Let us briefly consider the stability of these phases to time-reversal symmetric potentials, either commensurate with the lattice, or arising from quenched disorder. Focusing on commensurate potentials (of very small amplitude compared to the bandwidth), provided the separation between the bulk Weyl nodes is commensurate, any WSM phase can be gapped out in the bulk by an appropriate periodic potential. This will naturally gap out the corresponding surface Fermi arcs as well. Coming to the W2$'$ phase, such a periodic potential would gap out the Weyl nodes, but cannot destroy the Chern numbers of the fully gapped bands. Thus, we conclude that the W2$'$ phase is unstable to becoming a simple LCI. Similarly, the $I'$ phase becomes a trivial insulator when subjected to a small periodic potential of the appropriate period, which induces matrix elements between the counter-propagating surface modes and gaps them out.

Extending our discussion to quenched disorder, two lines of argument have been explored in the literature. It is known that the WSM is perturbatively stable to disorder in the renormalization group sense [26,36–42]. In this picture, the WSM undergoes a transition to a diffusive metal at a nonzero critical disorder strength. Along a different line of reasoning, however, the nonperturbative effects of large but rare Griffiths regions have been argued to destroy the WSM at arbitrarily weak disorder [43–46], and make it a diffusive metal at the longest lengthscales. If the first scenario prevails, the WSM phases uncovered in this work (and, significantly, the W2$'$ phase) will be stable to weak disorder. However, if the second scenario is proven to hold generically, the WSM phases and the W2$'$ phases will be destroyed immediately for arbitrarily weak disorder. This is because a diffusive metal will destroy the quantization of the

Hall conductance immediately. However, the LCI and the $I'$ phases, being fully gapped in the bulk, are expected to be stable to arbitrary weak disorder.

Let us comment briefly on potential experimental realizations of the physics uncovered here: There is a theoretical proposal for the simplest Weyl semimetal with only two Weyl nodes, based on inserting magnetic layers of Mn into a layered topological insulator such as $Bi_2Sb_3$ [47]. According to theory, the material should be a WSM with two Weyl nodes when the Mn layers are ferromagnetic [48]. Clearly, very large magnetic fields are needed to realize even $q \simeq 100$ in this realization. In a different direction, there are several proposals for realizing a Weyl semimetal in an optical lattice [49–51]. Very recently one such proposal has been also experimentally realized [52]. Since orbital fields can be imposed in a frequency driven optical lattice [53, 54], approaching commensurate fields seems more achievable in this realization.

Many open questions remain. An important one is the effect of interactions on the phase diagram. At $B = 0$, interactions have been conjectured to drive the WSM into numerous phases, including charge density wave [55–58], superconducting [59], Mott insulating [60], and even fractionalized [61, 62] phases. Renormalization group analyses have also been carried out keeping all symmetry-allowed interactions [63, 64]. As we have shown, near a quantum phase transition between different WSM phases, the spectrum near zero energy becomes quadratic in one direction, remaining linear in others. This would suggest a greater instability to certain types of interactions near the phase boundaries.

Our work could also be extended in a different direction. Our starting point in this study was a time-reversal broken WSM; adding the orbital magnetic field did not change this. However, we did find extra phases W2$'$ and $I'$ which appear only in the presence of a magnetic field. It would be of interest also to study time-reversal symmetric, inversion broken, Weyl semi-metal in a magnetic field, where the introduction of the magnetic field would introduce time-reversal breaking as well. We look forward to answering these and other questions in future work.

# Acknowledgements

AD was supported by the German-Israeli Foundation (GIF) Grant No. I-1505-303.10/ 2019 and the Minerva Foundation. AD also thanks Dean of Faculty fellowship Weizmann Institute of Science, Israel, Israel planning and budgeting committee, and Koshland Foundation for financial support. SR and GM would like to thank the VAJRA scheme of SERB, India for its support. GM is grateful for partial support from the US-Israel Binational Science Foundation (Grant No. 2016130) and the Aspen Center for Physics (NSF grant PHY-1607611) where this was completed.

# A   Symmetries of the lattice model

We have considered a minimal two-band lattice model of Weyl semimetals, Eq. 1, which breaks time reversal symmetry but keeps the inversion symmetry. Since the model involves pseudospin, the time reversal operation is just complex conjugation. For the case of isotropic hoppings, the symmetries of this model can be found in Ref. 65. Now let us look at in more detail at the crystalline symmetries which our anisotropic model has:

### A.1  Rotation:

It has a single two-fold rotation symmetry only about the $x$-axis and the symmetry transformation ($C_{2x}$) is given by

$$C_{2x} \, c_s(n_x, n_y, n_z) C_{2x}^{-1} = (\sigma_x)_{ss'} \, c_{s'}(n_x, -n_y, -n_z). \tag{13}$$

It can be easily checked that this symmetry operation leaves the Hamiltonian in Eq. 1 invariant i.e. $C_{2x} H C_{2x}^{-1} = H$.

### A.2  Mirror reflection:

The model is symmetric under the following mirror reflections about the $yz$ ($M_x$), $xz$ ($M_y$) and $xy$ ($M_z$) planes:

$$M_x \, c_s(n_x, n_y, n_z) M_x^{-1} = (\sigma_0)_{ss'} \, c_{s'}(-n_x, n_y, n_z), \quad M_x i M_x^{-1} = i\,, \tag{14}$$

$$M_y \, c_s(n_x, n_y, n_z) M_y^{-1} = (\sigma_x)_{ss'} \, c_{s'}(n_x, -n_y, n_z), \quad M_y i M_y^{-1} = -i\,, \tag{15}$$

$$M_z \, c_s(n_x, n_y, n_z) M_z^{-1} = (\sigma_0)_{ss'} \, c_{s'}(n_x, n_y, -n_z), \quad M_z i M_z^{-1} = -i\,. \tag{16}$$

All the above three symmetry transformations $M_x$, $M_y$ and $M_z$ leave the Hamiltonian in Eq. 1 invariant i.e. $M_\mu H M_\mu^{-1} = H$, $\mu = x, y, z$. We note that the symmetry operations given by $M_y$ and $M_z$ are not pure mirror reflections, rather they can be thought of as a product of mirror and time reversal operations. On the other hand, the two fold rotation $C_{2x}$ about $x$ axis and the mirror reflection $M_x$ about the $yz$ plane together give the inversion operation, and therefore the Hamiltonian is also inversion symmetric.

### A.3  Particle-hole transformation:

We find that the following particle-hole transformation, which also includes the mirror reflection about the $xz$ plan,

$$P_x c_s(n_x, n_y, n_z) P_x^{-1} = (\sigma_x)_{ss'} \, c_{s'}^\dagger(n_x, -n_y, n_z), \quad P_x i P_x^{-1} = i\,, \tag{17}$$

takes the Hamiltonian $H$ in Eq. 1 to $-H$ i.e. $P_x H P_x^{-1} = -H$. Therefore this symmetry forces the spectrum of $H$ to be symmetric about zero energy.

Now we check on which of these symmetries survive in the presence of an applied uniform external magnetic field along the $z$-direction. In presence of the magnetic field, the $x$-hopping $T_x = t_x \sigma_x$ picks up a phase factor $\exp(-i2\pi y \phi/\phi_0)$, which breaks all the symmetries, except the mirror reflection $M_y$ about the $xz$ plane (Eq. 15) and the particle-hole transformation $P_x$ (Eq. 17). Therefore the symmetry realized by $P_x$ survives even in the presence of the magnetic field, and ensures that the spectrum is symmetric about zero energy.

## B  Fermi arc surface states at zero and finite commensurate fields

The pseudospin polarization of the Fermi arc surface states changes its polarization character as we go from zero field to finite commensurate fields for both the WSM and the LCI phases. At zero field, the surface states are polarized in pseudospin along the $z$-direction. The right moving surface states of a slab finite along $z$-direction are localized at the top surface

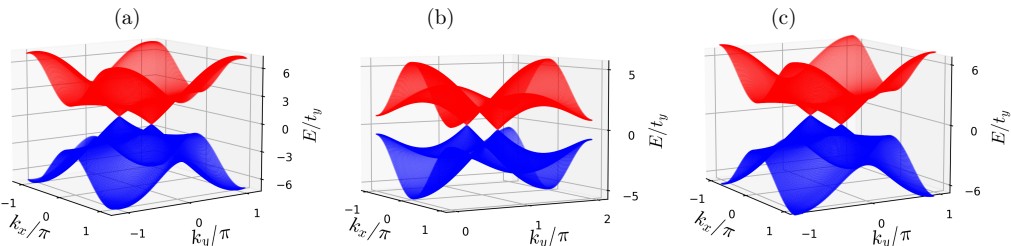

Figure 11: Energy spectrum of the zero field model showing Weyl nodes with linear dispersion along all directions near zero energy. (a) There are two Weyl nodes in the W2 phase which appear in the $k_z = 0$ plane. There exist four Weyl nodes in the W4 phase: two of them occur in the (b) $k_z = 0$ plane and the other two in the (c) $k_z = \pi$ plane

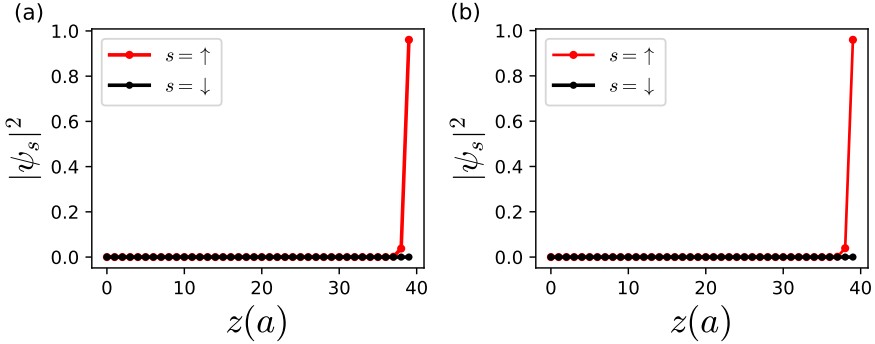

Figure 12: Eigenstate plots of the right moving Fermi arc surface states for the zero field ($\mathbf{B} = 0$) case for a finite slab along the $z$-direction with lattice sites $L_z = 40$. (a) This is in the LCI phase: the right moving surface states are localized on the top surface $z = L_z$ and are polarized along $z$-direction with pseudospin $s =\uparrow$. (b) This is in the W2 phase: the right moving Fermi arc states are localized on the top surface $z = L_z$ and they are polarized along $z$-direction with pseudospin $s =\uparrow$.

$z = L_z$ and they are polarized along the positive $z$-direction with pseudospin $s =\uparrow$. In contrast, the left moving surface states are localized at the bottom surface $z = 0$ and they are polarized along the negative $z$-direction with pseudospin $s =\downarrow$.

A few eigenstates plots are shown in Fig. 12 for this zero field case. When a magnetic field is switched on in the $z$-direction, surface states in the $k_x$-$k_y$ surface BZ of both the WSMs and LCI phases alter their polarization from the $z$-direction to the $x$-$y$ plane. Now the top surface hosts left moving surface states and the bottom surface hosts right moving surface states with amplitude being the same for both the $s =\uparrow$ and $s =\downarrow$ pseudospins. A few eigenstate plots are shown in Fig. 13 for non zero magnetic field with $q = 2$.

## C   Zeros of Bloch-Hofstadter Hamiltonian for arbitrary $q$

In this section, we will go through a detailed discussion of how we obtained the topological phase boundaries, the critical values and the phase diagram in the presence of an arbitrary commensurate magnetic field. We begin by writing the Hofstader Hamiltonian $h_\phi(\mathbf{k})$ in Eq. 6 as $h_\phi(\mathbf{k}) = \Psi^\dagger \, \tilde{h}_\phi(\mathbf{k}) \, \Psi$, with $\Psi = (\psi_\uparrow, \psi_\downarrow)^T$, where $\psi_s = \left(c_{0,s}(\mathbf{k}), c_{1,s}(\mathbf{k}), ..., c_{q-1,s}(\mathbf{k})\right)^T$ and

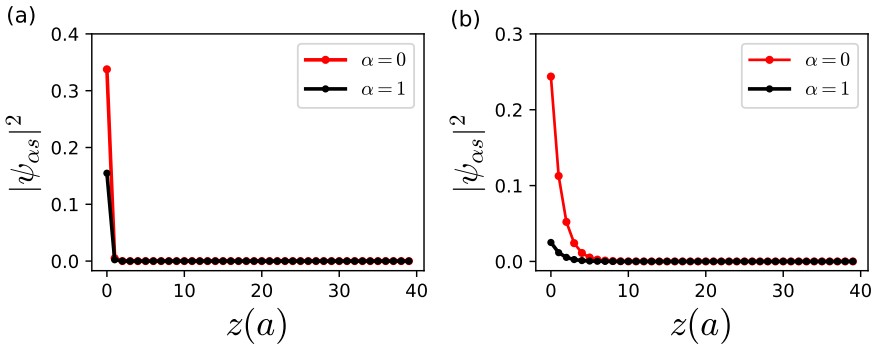

Figure 13: Eigenstate plots of the right moving Fermi arc surface states in the presence of a finite commensurate field ($B \neq 0$) with $q = 2$, for a finite slab along the $z$-direction with lattice sites $L_z = 40$. Here $\alpha = 0, 1$ refer to the coordinate of the lattice sites inside the magnetic unit cell. (a) This is in the LCI phase: the right moving Fermi arc states are localized on the bottom surface $z = 0$ and both the pseudospin components have the same amplitude. (b) This is in the W2 phase: the right moving Fermi arc states are localized at the bottom surface $z = 0$ and both the pseudospin components have the same amplitude.

$s \equiv (\uparrow, \downarrow)$. The matrix $\tilde{h}_\phi(\mathbf{k})$ is given by

$$\tilde{h}_\phi(\mathbf{k}) = \begin{pmatrix} \mathbf{A} & B \\ \mathbf{C} & \mathbf{D} \end{pmatrix}, \tag{18}$$

where all the blocks are of same dimension $q \times q$ and are given by

$$\mathbf{A} = -\mathbf{D} = 2t_z^{(2)} \sin k_z \mathbf{I}_q \quad \text{and} \tag{19a}$$

$$B = \mathbf{C}^\dagger = \begin{bmatrix} m_0 & -u & 0 & 0 & \ldots & -ve^{ik_y q} \\ -v & m_1 & -u & 0 & \ldots & 0 \\ 0 & -v & m_2 & -u & \ldots & \ldots \\ \ldots & \ldots & \ldots & \ldots & \ldots & \ldots \\ 0 & 0 & \ldots & -v & m_{q-2} & -u \\ -ue^{-ik_y q} & 0 & \ldots & \ldots & -v & m_{q-1} \end{bmatrix}. \tag{19b}$$

Here $\mathbf{I}_q$ is the identity matrix of dimension $q \times q$, $u = t_y^{(1)} - t_y^{(2)}$, $v = t_y^{(1)} + t_y^{(2)}$, $m_\alpha = f_1^\alpha(\mathbf{k}) = 2\left(M - t_z^{(1)} \cos k_z - t_x \cos\left(k_x + \frac{2\pi p}{q}\alpha\right)\right)$, and $\alpha \in [0, q-1]$ is the sublattice index. The eigenvalues $\lambda$ satisfy

$$\det \begin{bmatrix} \mathbf{A} - \lambda \mathbf{I}_q & B \\ \mathbf{C} & \mathbf{D} - \lambda \mathbf{I}_q \end{bmatrix} = 0. \tag{20}$$

Clearly the matrix $\mathbf{C}$ commutes with both $\tilde{\mathbf{D}} = \left(\mathbf{D} - \lambda \mathbf{I}_q\right) = -\left(\lambda + 2t_z^{(2)} \sin k_z\right)\mathbf{I}_q$ and $\tilde{\mathbf{A}} = \left(\mathbf{A} - \lambda \mathbf{I}_q\right) = -\left(\lambda - 2t_z^{(2)} \sin k_z\right)\mathbf{I}_q$. The left hand side of Eq. 20 is identically equal to $\det\left(\tilde{\mathbf{A}}\tilde{\mathbf{D}} - B\tilde{\mathbf{D}}^{-1}\mathbf{C}\tilde{\mathbf{D}}\right)$. Using the commutation of $\mathbf{C}$ and $\tilde{\mathbf{D}}$, Eq. 20 can be simplified to

$$\det\left(\gamma \mathbf{I}_q - \mathbf{B}\mathbf{B}^\dagger\right) = 0, \tag{21}$$

where we have used $\tilde{\mathbf{A}}\tilde{\mathbf{D}} = \gamma\mathbf{I}_q$, $\gamma = \lambda^2 - \left(2t_z^{(2)}\sin k_z\right)^2$ and $\mathbf{C} = B^\dagger$. So the eigenvalues are given by

$$\lambda = \pm\sqrt{\gamma + \left(2t_z^{(2)}\sin k_z\right)^2}. \tag{22}$$

The spectrum (we have suppressed the momentum argument $\mathbf{k}$ of $\gamma$ and $\lambda$ for notational simplicity) is obviously particle-hole symmetric, which is a consequence of the particle-hole symmetry $P_x$ of the Hamiltonian discussed in Appendix A. Eq. 21 implies that $\gamma$ is the eigenvalue of the positive semi-definite matrix $\mathbf{BB}^\dagger$ and so it must satisfy $\gamma \geq 0$. Therefore, for the spectrum to be gapless (which means $\lambda = 0$), we must have $\gamma = 0$ and $\sin k_z = 0$. The only possibilities are $k_z = 0$ and $\pi$ at which energy gap can close in the $k_z$ direction.

Since $\gamma = 0$ for energy spectrum to be gapless, we get the following determinant vanishing condition from Eq. 21,

$$\det(B) = 0, \tag{23}$$

where we have used the identity $\det(\mathbf{BB}^\dagger) = \det(B)\det(B^\dagger)$. To get the explicit gap closing condition in terms of the parameters of the theory, we must compute the determinant of the $q \times q$ matrix $B$. Looking at the matrix $B$, we find that it is particularly simple to calculate determinant of $B$ if either $u = t_y^{(1)} - t_y^{(2)}$ or $v = t_y^{(1)} + t_y^{(2)}$ is zero ($B$ then becomes almost a lower/upper triangular matrix). This can happen only when $|t_y^{(1)}| = |t_y^{(2)}|$. We choose $t_y^{(1)} = t_y^{(2)}$ ($= t_y$ say) such that $u = 0$ as discussed in the main text, and we use $t_y$ as a unit of energy. The other choice $t_y^{(1)} = -t_y^{(2)}$ which makes $v = 0$ will be equally valid and will not make any difference to our final results. Now we can expand the determinant of $B$ (setting $u = 0$) about the first row to get

$$\det(B) = \left\{\prod_{\alpha=0}^{q-1} m_\alpha\right\} + (-1)^{q+1}(-ve^{ik_y q})(-v)^{q-1}$$
$$= \left\{\prod_{\alpha=0}^{q-1} m_\alpha\right\} - e^{ik_y q}v^q. \tag{24}$$

For $\det(B) = 0$, the imaginary part $-v^q \sin qk_y$ and the real part must vanish. This implies that the energy gap can close only at $k_y = 0$ and $\pi/q$, along the $k_y$ direction (recall $0 \leq k_y \leq 2\pi/q$). For the real part of $\det(B) = 0$ to vanish we must have

$$\prod_{\alpha=0}^{q-1} m_\alpha = (-1)^\mu v^q = (-1)^\mu 2^q t_y^q, \tag{25}$$

where $\mu$ takes values 0 and 1 corresponding to the closing of the gap at $k_y = 0$ and $\pi/q$ respectively. We have used $v = t_y^{(1)} + t_y^{(2)} = t_y + t_y = 2t_y$ in the above expression. Recall that all the $k_x$ dependence is in the $m_\alpha = 2\left(M - (-1)^\nu t_z^{(1)} - t_x\cos\left(k_x + \frac{2\pi p}{q}\alpha\right)\right)$. Here $\nu$ takes values 0 and 1 which correspond to closing of the gap at $k_z = 0$ and $\pi$ respectively. Equation 25 gives us the final set of conditions which are to be satisfied by all the parameters of the theory for the energy spectrum to be gapless and thus completely determines the phase boundaries of the topological phase diagram.

It is possible to simplify the left hand side even further. We will use the following identity for the finite product [66]

$$\prod_{\alpha=0}^{q-1}\left(2g + 2\cos\left(k_x + \frac{2\pi p}{q}\alpha\right)\right) = 2\left[T_q(g) + (-1)^{p+q}\cos qk_x\right], \tag{26}$$

where $p$ and $q$ are relative primes and $T_q(g)$ is the Chebyshev polynomial of first kind of degree q. We can then simplify the left hand side of Eq. 25 and obtain the following,

$$\prod_{\alpha=0}^{q-1} 2\left( M - (-1)^{\nu} t_z^{(1)} - t_x \cos\left( k_x + \frac{2\pi p}{q}\alpha \right) \right) = 2(-t_x)^q \left[ T_q(g) + (-1)^{p+q} \cos q k_x \right], \quad (27)$$

where we identify $g = \left( -M + (-1)^{\nu} t_z^{(1)} \right)/t_x$. Rewriting Eq. 25 with the above simplification, we get the following expression for $k_x$

$$\cos q k_x = (-1)^{p+q} \left[ -T_q(g) + (-1)^{\mu-q} 2^{q-1} (t_x/t_y)^{-q} \right]. \quad (28)$$

Note that Eq. 28 involves only the two parameters $(g, t_x/t_y)$. For a given $(g, t_x/t_y)$, if $k_x = k_0$ is a solution of Eq. 28, then $k_x = -k_0$ too satisfies the same equation. Further, since $\cos q(\pm k_0 + 2\pi m/q) = \cos q k_0$, $k_x = k_0 + 2\pi m/q$, $m$ runs over $0, 1, 2, ..., (q-1)$; and $k_x = -k_0 + 2\pi m/q$, where now $m$ runs over $1, 2, ...., q$; are also solutions of Eq. 28. We recall that $k_x$ lives in the magnetic BZ, $0 \leq k_x \leq 2\pi$, and in the above $k_0$ is restricted to $0 \leq k_0 \leq 2\pi/q$. Hence, for a given $\mu$ and $\nu$, there are a total $2q$ number of $k_x$ values in the magnetic BZ where the spectrum is gapless. We have numerically verified that these distinct gapless points in magnetic BZ are the Weyl nodes in the theory.

We can also obtain the boundaries of the topological phases from Eq. 28. Note that the gapless point $k_0$ could be anywhere in between 0 and $2\pi/q$. But $\cos q k_0$ is distinct only in the range $0 \leq k_0 \leq \pi/q$. So when we vary $k_0$ from 0 to $\pi/q$, the Eq. 28 describes gapless regions in the parameter space. Solutions of Eq. 28 exist only if the value of the RHS lies in the interval $[-1, 1]$. For a given $\mu$ and $\nu$, the spectrum has $2q$ zeros (gapless points in BZ) when RHS $\in (-1, 1)$. As explained in the argument before Eq. 3, phase transitions, which imply a change in the number of solutions for a particular $\mu, \nu$, can occur only when the RHS is at the edge of its allowed range, namely $\pm 1$. Therefore the phase boundaries in the $(g, t_x/t_y)$ space are given by

$$-T_q(g) + (-1)^{\mu-q} 2^{q-1} (t_x/t_y)^{-q} = (-1)^{q+p}, \quad (29a)$$

$$-T_q(g) + (-1)^{\mu-q} 2^{q-1} (t_x/t_y)^{-q} = -(-1)^{q+p}. \quad (29b)$$

Eqs. 29a and 29b are obtained from Eq. 28 by setting the RHS equal to 1 and $-1$ respectively. Note the dependence of $p$ in the above equations; it only matters whether $p$ is even or odd. But even and odd values of $p$ merely interchange the equations Eq. 29a and Eq. 29b. So $p$ does not affect the topological phase diagram at all. Each of the above equations is a set of four equations, because both $\mu$ and $\nu$ take values 0 and 1 ($\nu$ enters through $g = (-M + (-1)^{\nu} t_z^{(1)})/t_x$). Note that the phase boundaries given in Eq. 29 are solely determined by only two parameters $(g, t_x/t_y)$. Therefore, the topological phase diagram is essentially controlled by the two parameters $(M/t_y, t_x/t_y)$. The role of the parameter $t_z^{(1)}$ which enters through $g$ is to merely shift the origin of $M$ - it does not lead to any new phase. The topological phase diagram is shown in Fig. 1. There are gapless WSM phases, the W2′ phase, a gapped Layered Chern Insulator (LCI) phase, an unusual $I'$ insulator phase along with a trivial insulator phase.

# D  Critical $M^c$ and $t_x^c$ for LCI phase

The critical values $M^c$ and $t_x^c$ which are defined in the main text in the Sec. 3.3.1 can be derived as follows. From the graphical visualization of the critical curves (phase boundaries) which can be seen explicitly in the figures in Fig.1 in the main text, we find that $t_x^c$ and $M^c$

are given by the intersection of the following two critical curves (obtained by putting $\mu = 1$, $\nu = 0$ in Eq. 29a and $\mu = 0$, $\nu = 0$ in Eq. 29b)

$$-T_q(g) + (-1)^{1-q} \, 2^{q-1} \, (t_x/t_y)^{-q} = (-1)^{q+1} \,, \tag{30a}$$

$$-T_q(g) + (-1)^{-q} \, 2^{q-1} \, (t_x/t_y)^{-q} = -(-1)^{q+1} \,, \tag{30b}$$

where, now, $g = (-M + (-1)^\nu t_z^{(1)})/t_x = (-M + t_z^{(1)})/t_x$ and we have used $p = 1$. We can easily find $t_x^c$ by subtracting Eq. 30a from Eq. 30b and the corresponding $M^c$ can be found by adding the two equations to get $T_q(g) = 0$ and solving for $M^c$. We find the following solutions for $q > 1$,

$$t_x^c = t_y \, 2^{1-1/q} \,, \tag{31}$$

$$M_c = t_z^{(1)} + t_x^c \cos(\pi/2q) = \left(1 + 2^{1-1/q} \cos(\pi/2q)\right) t_y \,. \tag{32}$$

For $q = 1$, we have $t_x^c = t_y$ and $M^c = 2t_y$ (where we have used $t_z^{(1)} = t_y$). In the $q \to \infty$ limit, these critical values approach to $M^c = 3t_y$ and $t_x^c = 2t_y$. This set of values gives the region $M > M^c, t_x > t_x^c$, where the LCI phase cannot appear.

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
