# Peer review of "Time-reversal-broken Weyl semimetal in the Hofstadter regime"

_SciPost Physics, doi:SciPost Phys. Core 5, 014 (2022)_

## Round 2 · Referee Report · Anonymous · 2021-10-1

Strengths
1. The paper is coherently written
2. The subject is timely
3. Results and discussions are overall sound
Weaknesses
1. The manuscript does not report much novelty
2. Taking into account the above not much beyond the phase diagram is addressed; the discussion of the phases is relatively shallow. This is also reflected in the motivation of the work
3. There is not much new physics reported, in particular no new observable effects or connection to materials
Report
In the submission by Abdulla et al. the authors consider a Weyl semi-metal [WSM] in presence of magnetic field. To this end they consider a rather well-known toy model for a WSM, eq 3. Adding the p/q rational flux induces a magnetic unit cell and a richer phase diagram with new phases dubbed W2' and I' for example. The phases and boundary states are considered numerically and, exploiting the simplicity of the model, also analytically in some cases.
The manuscript is coherently written and the question of considering topological phases in the presence of magnetism is timely. Although the interesting aspect of topological materials is that they can occur without external field, considering the effect of such a readily available experimental probe is interesting in its own right and the subject is increasingly receiving attention.
Nonetheless, I also have some concerns that I would like to discuss with the authors. Most importantly, there is the issue of novelty. There has been work on considering WSMs in magnetic fields that range from the weak field limit to strong field limit, see also ref 23 and quite some others. The authors say they go beyond these works by considering multiple Weyl nodes and different orientations of the magnetic field relative to the nodes. The point however is that having more Weyl nodes does not significantly change the physics of the nodes or total phase in itself. One of the earlier papers proposing WSMs [PRB 83, 205101 (2011)] in fact has many nodes, while the important local features per node [resulting in anomaly, stability of each node and their flux] can be effectively addressed by a “two-node” model akin to the one used. Possibly, the extra nodes can lead to more effects beyond just observing intermediate phases with more nodes, but then one really has to consider crystalline symmetries and their role to signify the physics. The authors neither give such crystalline interpretations nor motivate why their analysis involving extra nodes is important or worthwhile.
This rather shallow perspective is also illustrated in the superficial discussion of the I' phase in Sec 3.3.4; the authors merely state that the phase diagram features an insulating phase with counter propagating states. They rightfully connect this to surviving crystalline symmetries, but make no deeper analysis. Hence, not much is gained other then just identifying the phase in the phase diagram. Evidently, the model has an inversion symmetry and generalized mirrors that persist and should be taken into account. Such a broader analysis that includes the role of e.g. the inversion should be made. Similarly this deeper analysis should be related to the particle hole symmetry of the spectrum mentioned in section 3.1.
Overall, given the fact that this work is not radically new, at least its role as extension on previous results should be enforced by exploring all details in depth. Hence, these aspects should be improved.
Some other concrete points that I like to raise are
1. The authors say above Eq. 5 “...Note that we consider the orbital
effect only. The reason is the following: The orbital coupling, since it couples to the charge degree of freedom, is universal. On the other hand, the pseudospin label is a k-dependent linear combination of spin and orbital labels, which does depend sensitively on the microscopic material parameters...” I am confused by this. I agree that pseudospin label is k-dependent but the orientation can be important, although being model dependent. Indeed, consider for example the edge states. If it is spin polarised, meaning it is in the direction of one Pauli matrix, then anti-commuting Zeeman terms directly induce a gap, whereas commuting ones do not. Hence, there is an effect. This is rather similar to how the orbital parts also depends on specific model settings. This lack of broader view also ties to the motivation aspect discussed above.
2. In the discussion it is left implicit that when edge states are not mentioned periodic boundary conditions are used.
3. The authors say “...We find that the topological phase diagram, in general, depends only on three of the hoppings (tx,t(1)y ,t(1)z ) and the onsite mass parameter M...” Given that the phase diagram is determined essentially by the tuning of the mass it seems that one can scale out t_x and consider (1, t_y/t_x, t_z/t_x, M/t_x) as is usual for such kinds of topological toy models.
Finally some minor points
-The authors comment on the chiral anomaly in the second paragraph. The transport properties they describe semi-classically leads to the chiral anomaly, hence I would not use “due to” rather it is the other way around; the anomaly arises due to non-conservation of the density of states at each Weyl node.
-Ref 25 studies Chern and fragile insulators rather than Weyl or double Weyl. The se systems are then linked to higher order topology in contrast to ref 24.
Author: Faruk Abdulla on 2021-10-12 [id 1840]
(in reply to Report 1 on 2021-10-01)
First we would like to thank the referee for the report. The referee feels the work is coherently written and timely, but has some general questions regarding the novelty of the work, and more specific questions regarding particular statements made in the manuscript.
Novelty: There has been previous work exploring the effects of an orbital magnetic field in Weyl semimetals (WSMs). In particular, Ref 23 focused on issues concerning the chiral anomaly, and its signatures in a lattice model with only two Weyl nodes. In order to address the chiral anomaly, the B field has to have a component along the line joining the Weyl nodes.
Our work explicitly orients the field perpendicular to the line joining the Weyl nodes. Our focus is quite different from that of Ref. 23. We explore the entire phase diagram of a two-band lattice model with minimal crystalline symmetry in a commensurate orbital flux (p/q flux quanta per unit cell). There are five independent hopping parameters and one ``mass" parameter in the Hamiltonian. Setting one of them to unity still leaves five independent parameters.
Our work is the first to present even the zero-B phase diagram of this model in its full generality. What seems to have been missed in previous work (which imposed crystalline symmetries that reduced the set of hoppings) is that this two-band model at B=0 can exhibit a plethora of phases, including WSMs with two, four (PRL 115, 246603 (2015); PRB 93, 075108 (2016); PRX 8, 031076 (2018)), six and eight Weyl nodes, in addition to the layered Chern insulator which was pointed out in earlier work (above mentioned papers). So the occurrence of phases with large numbers of Weyl nodes is simply a consequence of reducing the symmetry of the model. It is a feature of our work, but not its focus.
Our main focus concerns the evolution of the phase diagram of the model with the fewest crystalline symmetries all the way from strong fields to the semi-classical limit. It is extremely helpful that we can obtain the entire spectrum and the phase boundaries analytically for arbitrary q (1/q flux quanta per unit cell), because it enables us to go to extremely large q, thus extremely small fields. The semi-classical limit is highly nontrivial in certain ranges of parameters, because for any nonzero B, the system can be in the fully gapped layered Chern insulator, while for zero B it is in a WSM phase. We elucidate the semi-classical limit by studying the spectrum, especially the gaps, as q becomes very large. In addition, we find a new insulating phase with a pair of non-Kramers counter-propagating modes on the surface.
In our opinion, there are quite a few new results in the manuscript.
Let us turn to the more specific criticisms of the referee.
-
Ignoring the Zeeman coupling. We disregarded it because spin-orbit coupling renders the Zeeman coupling non-universal. It also does not change the qualitative behaviour of either the bulk or the edge states. The referee gives the example of the Zeeman coupling being important for degenerate edge states. We agree that if degenerate states are involved, one needs to be careful about ignoring the Zeeman coupling. However, for our model, the Zeeman coupling makes no qualitative difference. First let us consider bulk states, which can be degenerate at a Weyl node. Adding a k-dependent Zeeman coupling will merely shift the Weyl node to a slightly different location. So adding a k-dependent Zeeman coupling can certainly make quantitative changes to the bulk phase diagram, but cannot alter it qualitatively. Next, let us consider the surface states. The WSM, layered Chern insulator, and I' phases all have non degenerate Fermi arc-like surface states. Thus, the Zeeman coupling can only shift the locations of the surface states, but cannot gap them out, or make any other qualitative difference. Perhaps we did not explain this clearly enough in the text, and will endeavour to do so once all the referee comments are in.
-
Periodic boundary conditions are used when studying bulk spectra, which we believed to be customary, and not worth specific mention.
-
Scaling out one of the hopping parameters: As already noted, there are six independent parameters. We set three of them equal (see caption of Fig 1). We emphasize that this does not increase the symmetry of the model, so the model still has minimal symmetry. We then use one of these hopping parameters as our unit of energy. This leaves two parameters, in terms of which we plot the phase diagram. Thus our main phase diagram is not ``determined essentially by the tuning of the mass".
-
Symmetry analysis of the I' phase. For nonzero B we find a new insulating phase I' in certain regions of the phase diagram. The surface modes of I' occur only on some surfaces, and consist of a pair of counter-propagating modes separated in momentum. In the clean crystal, the modes are protected by translation symmetry.
Recall that we deliberately chose to have as little symmetry in the two-band model as possible in order to explore the general phase diagram. At B=0, our model has a $C_2$ rotation around x, a mirror symmetry of reflection in the yz plane, and hence also the inversion symmetry. Once an orbital flux is turned on, the model loses all these symmetries. Of course, translation symmetry remains since we do not consider disorder.
Since the model has only translation symmetry for nonzero orbital flux (which is our focus) we did not include a detailed symmetry analysis. We will update the section on the I' phase with a more complete symmetry analysis once all the referee reports are in.
The referee correctly pointed out that the Ref 25 is not about Weyl and double Weyl. We will place this citation appropriately once all the referee reports are in.
Author: Faruk Abdulla on 2022-01-12 [id 2096]
(in reply to Report 3 on 2021-11-02)We are grateful to Referee 3 for an extremely careful and detailed reading of the manuscript, and for their constructive comments which have made the manuscript much better. Below, we address the points raised by the referee in sequence.
1) The referee suggests that we make the abstract much more specific and present our results in summary there. We have done so, keeping in mind the limited space available.
2) The referee suggests that we sharpen the description of our work in the introduction, and present the context as well. We have done so to the best of our ability.
3) The referee would like to see not only the Fermi arcs themselves, but also more detail on how they decay into the bulk. We have added four new sub-diagrams to Fig. 3, and radically updated Fig. 4 to clarify this issue.
4) The results of arXiv:2108.05562, mentioned by the referee, does superficially resemble our W2$'$ phase, with its coexistence of Weyl bands and LCI bands. However, we believe the origins of their results are very different from ours. To be specific, the restructuring of Fermi arcs in arXiv:2108.05562 is connected to the higher order charge of the Weyl nodes and to the existence of higher order hinge-states which require crystalline symmetry to stabilise both the phases. They also find a phase where these two types of surface states coexist. The Fermi arc states due to the bulk Weyl nodes of higher order charge exist for many different surface orientations, but the hinge states due to the higher order topology exist only for a particular surface orientation. On that particular surface orientation, the higher order Fermi arc states and the hinge states join. By contrast, our model has only simple Weyl nodes, no higher-order topology, and minimal crystalline symmetry. We have referred to the preprint and added a brief comment on the differences in Section 3.3.3, where we discuss the W2$'$ phase.
5) We have now reworked the summary, adding some of the important references concerning interactions in bulk WSMs.
6) We have also modified the discussion of the effects of disorder on WSMs, presenting the two different points of view on the evidence so far.
7) We have corrected the identical citation issue.
8) In the summary, we have discussed the possibility of realizing models of this type in real materials as well as in optical lattices, along with appropriate references.

---

## Round 2 · Referee Report · Anonymous · 2021-10-15

Report
In this paper, the authors do a detailed analysis of the phase diagram of a Weyl semimetal in the presence of large magnetic field. They find a handful of new features, the most notable of which in my mind is the phase they label W2’, which corresponds to coexisting Weyl nodes and layered Chern insulator. I think the work is correct so merits publication, but does not currently rise to the level of importance that Scipost is targeting.
With papers such as this, there are a few questions that I ask to determine whether the paper has novel results, namely:
1. Are there fundamentally new phases of matter? If so, is there any meaningful discussion of the physical import of the new phase of matter (experimental detectability, topological response, etc.)
2. Is there a proposal for how this model Hamiltonian may be realized experimentally?
3. Is there a meaningful discussion of stability to, e.g., disorder or interactions?
Of these three topics, I only see discussion of the third. Without some reason to believe this is more interesting than stated (via criteria 1 and 2), I recommend against publication in SciPost.
Author: Faruk Abdulla on 2021-10-30 [id 1891]
(in reply to Report 2 on 2021-10-15)
First we would like to thank the referee for the comments. We are sorry to see that referee thinks our work is not suitable to Scipost. Below, we rebut the criticism in detail.
1) Novelty: We have answered this in our response to referee 1. We reiterate them here.
(i) We have solved the zero-field problem in full generality, without assuming any additional lattice symmetries other than inversion. Thus, our phase diagram includes the full set of phases possible in this model, with previous work finding only certain subsets of phases.
(ii) One of our major goals was to find evolution of the full phase diagram as we change the magnetic field which is known to be an unsolved problem to the best of our knowledge for such a model with minimal lattice symmetry. Our analytical solution helped us to construct the phase diagram for arbitrary flux of $1/q$ through the 2d unit cell. This allowed us to take the weak field limit $1/q \to 0$ which in turn allowed us to make contact with the semi-classical continuum approach. This has not been done before, to the best of our knowledge.
(iii) Experimental detection and topological response: One of the phases we find, W2$'$ is an interesting phase that has not been seen before. The coexistence of bands supporting a layered Chern insulator, and other bands supporting a Weyl semimetal, leads to a unique Hall response. Both the layered Chern insulator and Weyl semimetal's edge modes will contribute to the Hall conductivity. The 2d effective Hall conductance $\sigma_{yz}$ changes in a predictable way as we change different parameters of the theory including the magnetic field. We plan to add a brief explanation and update the manuscript when all the referee reports are in.
2) Experimental model Hamiltonian: There is a theoretical proposal for the simplest Weyl semimetal with only two Weyl nodes, based on inserting magnetic layers of $Mn$ into a layered topological insulator such as $Bi_2Sb_3$ (Nature Physics 5, 438 (2009)). According to theory, the material should be a WSM with two Weyl nodes when the $Mn$ layers are ferromagnetic (PNAS, 117, 27224 (2020)). To the best of our knowledge, it has not yet been experimentally confirmed that this is indeed such as WSM. In a different direction, there are several proposals for realizing a Weyl semimetal in an optical lattice (Scientific Reports volume 6, Article number: 33512 (2016); PRL 114, 225301 (2015); PRA 93, 063606 (2016)). Very recently one such proposal has been also experimentally realized (Science, 16 Apr 2021, Vol 372, Issue 6539).

---

## Round 2 · Referee Report · Anonymous · 2021-11-2

Strengths
Pedagogical technical discussion
Weaknesses
Lacks the vision of broad implications
Report
In this manuscript Abdulla et. al. study a two-band model for Weyl semimetals in the Hofstadter regime by threading a rational p/q flux quanta through each plaquet. This paper has already been reviewed by two other referees, who raised some concerns regarding the novelty, broad implications of this study and the experimental feasibility of the model the authors study. One referee in particular raised some genuine technical questions. Even though I find the responses from the authors to be reasonably convincing, I leave it up to the respective referee to judge the correctness of the responses. Given that two referees asked quite reasonable and probing questions, I will comment on the suitability of this article for publication in SciPost. First of all, I find this work to be convincing and the results to be appealing to a set of audience working on Weyl semimetals, which nowadays is fairly large. Therefore, this paper can be published in SciPost. However, prior to publication, I want the authors to take into account the following suggestions, which in my opinion should increase the impact of this article.
1. I strongly recommend the authors to expand the Abstract of this article. For example, the last sentence from the Abstract is pretty vague and it only mentions what calculations the authors do, while being completely silent about the outcome. I understand that an Abstract cannot contain every technical detail and outcome. But, authors should also bear in mind that nowadays often readers have time to read the Abstract of an article before delving into the paper itself. Therefore, Abstract needs to be punchy and must bear a flavor of broad appeal to attract more audience. Some of the authors are senior and established scientists. So, they do not need my guidance to improve an Abstract. Nonetheless, I strongly urge the authors to extend and polish the Abstract.
2. In the same spirit, I strongly urge the authors to improve the presentation of the Introduction. The first half of the Introduction reads very nicely, providing a broad overview of the field of Weyl semimetals. However, the second half reads very rushed to me. In principle, the authors can divide the whole Introduction into a few parts, for example (a) general Introduction, which is already there, (b) an extensive summary of their results (this is the part I found to be rushed, thus needs more serious work), which should include references to appropriate figures from the main text and (c) broad impact of this work in the general field of Weyl semimetals or even better if the authors can connect it to topology in gapless systems. There are a few sentences in the Introduction which simply do not serve any purpose. For example "Many features of the phase diagrams are universal and independent of p". Statements like this need to be expanded and authors need to state what are the universal features they are referring to, for example.
Therefore, in brief the Abstract and Introduction require serious restructuring. Notice that SciPost is a very good journal and we collectively should maintain its high standard.
3. Fig. 2 for example the authors show the energy spectra for the surface states, which I believe correspond to the Fermi arcs. Instead of only showing the energy spectra, the authors should also display the localization of the Fermi arcs, especially how they become delocalized as the Weyl nodes are approached, as shown in PRB 96, 201401 (2017).
4. The distribution of the topological charges to the Weyl nodes and their connectivity via Fermi arcs, shown in Fig. 6, closely resemble intertwined Weyl phases, discussed in arXiv:2105.08443. I recommend the authors to consult this preprint and make connections, if they exist.
6. In the second last paragraph of the "Summary and Outlook" authors mention effects of electronic interactions in Weyl systems. But, I find it to be quite surprising that the entire paragraph does not contain a single reference, while ample efforts have been invested to investigate interaction effects in Weyl materials. For example, see PRB 87, 161107 (2013), PRB 90, 035126 (2014), PRB 95, 201102 (2017). The authors should do literature surveys and find other important works in this direction. But, this paragraph definitely demands at least a few key references.
7. The discussion on the disorder effects on Weyl semimetal is somewhat biased toward rare region effects and the authors cite multiple papers from a single group of researchers, which I find to be somewhat unfair. Let me first comment on physics. Authors should note that numerical evidence for the rare region effect is strictly slim, as rare states have only been observed in the very close vicinity of the WSM-metal critical point, roughly about 10-15% of the critical disorder strength. Even though it has been claimed that those rare states persist all the way down to infinitesimal disorder, there exists no real trend or direct evidence supporting such a claim. Given that authors have not studied the disorder effects by themselves, I suggest that they only state the current status of this problem, as the jury is still out. Also it is not clear yet whether those rare states really convert the system into a metal or not. Only if those rare rares lead to finite dc conductivity at infinitesimal disorder and at zero temperature the phase can be called a metal, according to the definition by Nevil Mott. Also they need to acknowledge efforts from other research groups on this problem by referring to the following articles PRL 112, 016402 (2014), PRL 113, 026602 (2014), PRB 93, 210302 (2016), PRB 94, 220201 (2016), ... I urge authors to expand the reference listing on this topic to honor the effort invested by different research groups.
8. A minor point: Ref. 28 and Ref. 36 are identical. Please fix it.
To summarize, I think this paper can be published in SciPost once the authors fix the issues I pointed out, and answer the questions two other referees asked. As far as the experimental relevance of this model is concerned, I suggest authors add more concrete discussion rather than saying "it can be realized in cold atomic setups".
PS: I want to say sorry to the authors for the delayed report. There were some personal issues, which I had to deal with and which I could not avoid.

---

## Round 3 · Referee Report · Anonymous (Referee 1) · 2022-1-27

Strengths

  • improved clarity
  • better storyline than before

Weaknesses

Inherent to the paper is a very narrow focus and lack of new insights

Report

I thank the authors for their reply.

The manuscript has improved and in principle it can be endorsed as a result.

Overall the response to my questions was satisfactorily. I however repeat that there are not many new insights. With my previous questions on novelty and motivation I intended to spark a discussion on this topic. The authors are correct that they analyze a slightly different problem compared to the references in the introduction. This was however not the idea of my comment. The point is that the authors consider a simple toy model for Weyl nodes [[in fact so simple that not all symmetries can be broken, due to presence of 2-band nn model]] and the introduction of the field not suprisingly invokes Weyl nodes and transitions. Such consequences of this effective Weyl model have been well considered in a very vast body of literature. So given that new general insights/principles are obtained, such a consideration is what it is, but at least narrow in terms of impact and new insight or applicability to experiment. Indeed also the crystalline phase is inherent to the toy model and for experiment will need the right symmetries and will suffer from the complication that this effective model for the Weyl nodes at the Fermi level will not capture any realistic system, especially in terms of edge states.
  • validity: ok
  • significance: ok
  • originality: low
  • clarity: high
  • formatting: perfect
  • grammar: perfect

Author:  Faruk Abdulla  on 2022-02-02  [id 2147]

(in reply to Report 1 on 2022-01-27)
Category:
remark
reply to objection

We thank the referee for finding our manuscript to be improved with better presentation. However, in the referee's opinion, there is not enough novelty in our paper. Furthermore, the referee feels that the model is too simple to be applied to experiments. While it is difficult to argue with an opinion, we respectfully disagree, and offer the following rebuttal.

  1. It is true that our model is simple, but we consider this a virtue. It is often easier to see qualitatively new results in a simple model than in one with all the complexities of a particular experiment. The simplicity of the model allows us to fully characterize the phase diagram for arbitrary commensurate fields, which is itself a new result.

  2. We also find two new phases, one which displays the coexistence of Weyl nodes and attendant surface Fermi arcs, and the surface conduction of a Layered Chern insulator. The only other example of this phenomenon currently known to us, arXiv:2105.08443, needs special symmetries. This is new and has direct experimental consequences.

  3. The fact that we can analytically find the phase diagram allows us to approach the semiclassical limit systematically, which is a new result.

  4. We are not quite sure what the referee means when they claim that the model is too simple to break all the symmetries. We have in fact considered the case with minimal lattice symmetry. We therefore expect the results to be more generic than those obtained in a model with a higher symmetry.

We appreciate the comments by the referee and would like to point out that referee three in the previous round found our results appealing.

---

## Round 3 · Referee Report · Anonymous (Referee 1) · 2022-2-3

Strengths

see report

Weaknesses

see report

Report

I thank the authors and already said the results are what they are and that I can endorse publication as reference for a calculation involving this model. However for the record I do like to state the following;

  1. We agree that the simpler the model the better to illustrate the physics. So indeed the toy model for 2 nodes in the bulk has since the Nielsen-Ninomiya work, https://www.sciencedirect.com/science/article/pii/0370269383915290, been very important, but such a toy model should stand at the basis of a universal new effect, e.g. Weyl excitations giving rise to a chiral anomalous effect. Just calculating the phase diagram of a toy model is not a result that induces much novelty. Furthermore the effects the authors discuss do start from a model that needs to match the experimental situation [this model of nodes at the Fermi level, these symmetries] and the surface cut, see also point 4.

  2. Here I like to point out that quite generally the relation to stacked Chern insulators and thus TRS breaking slice in momentum space [c.f. Also role of magnetic field as we know from e.g. Haldane model that TRS breaking terms can induce Chern layers] is rather well known. Indeed it is a manner to visualize Weyl states see Fig 1 in https://www.sciencedirect.com/science/article/pii/S1631070513001710, so when under magnetic field when one gets transitions the results are to be expected from these general point of view. The authors point out arXiv:2105.08443 but there is quite extensive literature on Weyl semimetals in magnetic fields, and with regard to their comment on the crystalline symmetry I refer to point 4 and 5.

  3. We agree about this but this is also tied to simplicity of the model.

  4. The definition of minimal crystalline symmetries is space group 1, i.e. P1 having no inversion [P1'] or two fold rotations. Due to the simplicity of the model [only nn hopping, 2 band] this is not realized and should be taken into account with their own comments and comparison to other papers in point 2, see my point 5.

  5. Already from a simple point of view it is puzzling why the authors claim novelty or minimal crystalline symmetries. Indeed, not wanting to start a discussion of how it is not surprising that a 2-band Weyl model will transition of course via Weyl points to possible layered Chern insulators etc or other Weyl phases or gapped phases, we can consider the authors' own reasoning in point 2. They state that arXiv:2105.08443 is different as it needs spatial symmetries. But if their system had no crystalline symmetries or only translations how is it possible that they find a gapped phase with protected edge states? Indeed, this is because of point 4. So discarding the fact that general reasoning shows that the authors' results are not surprising, even in their own context their arguments are confusing.

---

## Round 3 · Author Response

We thank all the referees, especially referee 3, for their detailed and contructive comments which have helped us to vastly improve the manuscript. We have extensively revised the introduction and the summary, by making more explicit the novelty and context of our work, emphasized the new phases found, and discussed their implications in detail.

---

## Round 3 · List of Changes

1. We have completely revised the abstract, making sure that new results are given due prominence.

2. We have thoroughly reworked the introduction in accordance with the suggestion of Referee 3. Now we provide a detailed explanation of the precise way in which our work extends previous results (minimal crystalline symmetry, field perpendicular to the line separating the Weyl nodes), and the importance of the semiclassical limit. We also list the phases we see, including the W2’ and I’ phases which are only seen at nonzero field.

3. Despite the minimal crystalline symmetry, there are a number of symmetries enjoyed by the zero-field Hamiltonian. For completeness, we have devoted an appendix to listing all these symmetries.

4. In accordance with Referee 3’s suggestion that we show the reader how the surface states decay into the bulk for the Fermi arc states and the surface states corresponding to the LCI, we have added four panels in Fig. 3, and an entirely new Fig. 4.

5. In subsection 3.3.3, we have added a paragraph to describe a topological response of W2′ phase, which is additive between the WSM surface states and the LCI surface states.

6. We have revised 3.3.4 describing the I′ phase. We offer a natural way to think about it as arising from the W2’ phase, to which it is always adjacent in the phase diagram.

7. The panels in Fig. 6 have been reorganised for clarity of presentation.

8. Panel 9c has been updated, and now shows the gap rather than its logarithm.

9. Two new references (33 and 34) concerning the annihilation of Weyl nodes in a magnetic field have been added in section 5.

10. In accordance with Referee 3’s suggestion, in the summary, the paragraphs about disorder and interactions have been completely revised, and references added.

11. A short paragraph on potential ways of experimentally realizing the model we use has been added in the summary.

---

## Editorial Decision

published